# Formalized aspect-oriented misuse case for specifying crosscutting security threats and mitigations

**Shumaila Iqbal**[1], **Rizwan Bin Faiz**[2], **Muhammad Usman**[3], **Shafiq ur Rehman**[4]*

1 Department of Computing, Riphah International University, Islamabad, Pakistan, 2 Department of Computer Science, Fazaia Bilquis College of Education, PAF Base Nur Khan, Air University, Rawalpindi, Pakistan, 3 Software Quality Engineering and Testing (QUEST) Laboratory, Islamabad, Pakistan, 4 College of Computer and Information Sciences, Imam Mohammad Ibn Saud Islamic University (IMSIU), Riyadh, Kingdom of Saudi Arabia

* srehman@imamu.edu.sa

## Abstract

Software applications are essential for managing daily life activities, including social interactions and business transactions, that significantly increase the need for security in sharing sensitive information. Misuse case modeling is used for identifying and analyzing security requirements in software applications. However, security threats and their corresponding mitigations are inherently cross-cutting concerns. These concerns are scattered and tangled within multiple functional requirements and cannot be modularized using traditional object-oriented techniques. The realization of misuse cases causes crosscutting threats and corresponding mitigations to be scattered and tangled across use cases, resulting in ambiguity, incomplete understanding, and insufficient analysis of security requirements. This study proposes a misuse case modelling method called Aspect-oriented Formalized Misuse Case (AFMUC). It specifies crosscutting security threats separately as an aspect misuse case and integrates them with use cases using an aspect-oriented approach. AFMUC provides structured guidelines and restriction rules for modeling crosscutting security threats and corresponding mitigations using aspect-oriented constructs such as Pointcut, Joinpoint Advice, and Introduction. The aspect threat model is then woven into the base use case model. Similarly, an aspect mitigation model is proposed to specify crosscutting mitigations following the AFMUC restriction rules. The aspect mitigation model is then woven into the base misuse case model. The proposed approach is applied to a case study and evaluated through a controlled experiment involving twenty-four students with a background in information security. The findings indicate that the AFMUC approach is practical and unambiguous for specifying and analyzing crosscutting security requirements. However, some aspect-oriented modeling constructs and restriction rules have been misapplied by students. This shows that while students favored the AFMUC approach, they may have found it challenging to

**Data availability statement:** All relevant data are within the manuscript.

**Funding:** This work is supported and funded by the Deanship of Scientific Research at Imam Mohammad Ibn Saud Islamic University (IMSIU) (grant number IMSIU-DDRSP2504).

**Competing interests:** The authors have declared that no competing interests exist.

apply the aspect-oriented constructs and restriction rules due to a limited exposure to aspect-oriented modelling.

## Introduction

Software applications are essential for various purposes such as information sharing, business transactions, shopping, banking, social interactions, and managing edge intelligence for real-time data processing and decision-making applications. This rapid use also increases the risk of security threats. That makes applications an attractive target for attackers, which leads to critical security breaches, including unauthorized access, information exposure, and others [1]. The primary factor contributing to these vulnerabilities is the presence of security flaws during the development process of software applications. It is evident that amongst all the identified security threats, 97.5% of threats exist only in the requirement phase of the software development life cycle as compared to other phases. The software applications are vulnerable to attack due to an insufficient analysis of the security threats [2].

Misuse case modelling [3] is a well-known threat modelling approach for security analysis early in the requirements phase of software development life cycle. It has emerged as a simple and effective approach to identify potential security threats within the functional requirements. As a result, it is a commonly used approach among threat modeling approaches like STRIDE, DREAD, and others [4–6]. Misuse case modeling consists of diagram and specifications, built on the standard use case modelling. It is used to analyze security threats and their mitigations. The misuse case specifications complement the misuse case diagram by providing a textual description of the threat scenarios, explaining *how a misuser could compromise the application's security* while attacking the functionality, and mitigation explains *how security threats are mitigated to protect the application* [7,8]. The security threats and their corresponding mitigations are defined as crosscutting concerns within the application. Crosscutting concerns are those aspects of an application that cut across multiple functional modules and cannot be encapsulated within a single module. These concerns are often scattered across various modules and tangled within a single module of an application. Similarly, crosscutting security threats threatens several functional requirements, causing threat scattering, and single functionality is threatened by several threats, causing threat tangling. And their corresponding crosscutting mitigations often mitigate multiple security threats, causing mitigation scattering, and a single threat is mitigated by various mitigations, causing mitigation tangling [9].

Misuse case model present these threats and mitigations separately, but this separation is not maintained in their realization during software development phase [10]. When realizing misuse cases, a threat can attack the attack surfaces (i.e., vulnerable points that expose the application to threats) of several use cases, causing *threat scattering*. Also, a single use case may contain fragments of several threats, leading to *threat tangling*. Similarly, in mitigation realization, a single mitigation prevents many threats while targeting the mitigation surfaces (i.e., threat points that need to be mitigated) in misuse cases, causing *mitigation scattering*. A single threat can

be protected by many mitigation cases, causing *mitigation tangling*. The existing misuse case specifications [3,7,8] lack in addressing the issues of scattering and tangling (i.e., specifying crosscutting security threats and their crosscutting mitigation). It causes ambiguity and misunderstanding of the threats and their mitigation, leading to an insufficient security requirements analysis.

Inspired by an Aspect-Oriented Modelling (AOM) approach [9], this study integrates the aspect-oriented constructs in misuse case specifications to develop a structured and formalized misuse case specification template called Aspect-oriented Formalized Misuse Case (AFMUC). AFMUC consists of AOM constructs and restriction rules such as pointcut, joinpoint, advice, and introduction, to specify the crosscutting security threats and corresponding mitigations through misuse case threat modelling approach. The objective is to resolve the problem of tangling and scattering while specifying the crosscutting security threats and their crosscutting mitigation in misuse cases. For elicitation and analysis in a structured and analyzable form, a better variant of misuse case modelling, i.e., Restricted Misuse Case Modelling (RMCM) [7], is adopted. The RMCM is based on a template and restriction rules, providing precise and unambiguous specifications of security threats and their mitigations. The RMCM template is extended with AOM constructs and new restriction rules targeting the unambiguous modelling of the crosscutting threats and their crosscutting mitigations. The AFMUC template presents the AOM constructs, (i.e., *pointcut*, *joinpoint*, *advice,* and *introduction)* and restriction rules to specify the crosscutting security threat as an *aspect threat model* and their *aspect mitigation model* in an unambiguous and analyzable manner.

The proposed AFMUC approach is applied to a case study to assess its applicability, and a controlled experiment is conducted to empirically evaluate the ambiguity in terms of understandability, applicability, restrictiveness, and error rate. The results show that the approach unambiguously specifies the crosscutting threats and their crosscutting mitigation. Unlike the traditional misuse case approach, AFMUC is a novel approach and unambiguously identifies the specific vulnerable points within use cases (precondition, basic flow, postcondition). Furthermore, it supports precise and corresponding mitigation by defining the specific mitigation points for countermeasures within the precondition, basic threat flow, and postcondition of misuse cases. Further, the proposed approach benefits the security requirement engineers in addressing crosscutting threats and their mitigation early in the requirement phase. This facilitates a comprehensive and common understanding of threats and their mitigations, ensuring the consistent development of secure applications throughout the software development life cycle. It also guides the security testers to get detailed information about the attack surfaces and the mitigation surfaces, helping them to identify where to focus their efforts.

The main contributions of this paper are as follows:

1. The AFMUC, a misuse case description template with AOM constructs and restriction rules, is developed to specify crosscutting threats and their mitigations.

2. The ambiguity of the proposed approach is evaluated through a controlled experiment.

3. The applicability of the proposed AFMUC approach is assessed by applying it to a case study.

4. The results show that the AFMUC approach is unambiguous in terms of understandability, applicability, and restrictiveness. A relatively high error rate is observed while applying pointcut, joinpoint, and advice (Before, After, and Around). This indicates that although participants understood the underlying AFMUC rules, they faced difficulties in implementing them. It is believed that these difficulties may stem from their limited background in aspect-oriented modeling. These insights necessitate training that focuses on these AFMUC constructs and restriction rules.

The rest of the paper is organized as follows: The "Related Work" section shows an overview of the previous relevant studies. Then the "Overview of the Proposed Approach" section describes the proposed AFMUC template and discusses the details of the aspect-oriented constructs and restriction rules specific to the aspect misuse case threats and aspect mitigation cases. The "Example Case Study" section focuses on how to apply the proposed AFMUC approach to a case

study. The "Empirical Evaluation" section represents the details of the controlled experiment to assess the ambiguity of the proposed AFMUC approach. This section presents research questions, hypotheses, the design process, and the evaluation instrument of the experiment. The "Results and Analysis" section discusses the results of the experiment as well as the application of the proposed approach. The "Discussion" section presents the potential implications of these findings, comparison to other existing threat modelling approaches, and the limitations of the proposed AFMUC approach. The "Threats to Validity" section elaborates on the potential threats, then the "Conclusion" section concludes the paper. Lastly, open research directions of this study are highlighted in the section "Future Work".

## Related work

Modelling crosscutting security concerns using the UML model with an aspect-oriented approach is a significant area of study [11]. Literature shows that the AOM constructs (i.e., pointcut, joinpoint, advice, and introduction) have been integrated into various UML diagrams, such as AspectSM [12] and Aspect Sequence diagram [13] are effective in modelling crosscutting security threats. To the current understanding, there is a lack of literature specifically addressing crosscutting security concerns in the context of misuse case modelling. This issue, however, is addressed with use cases. Existing methodologies predominantly employ aspect oriented modelling to address crosscutting security concerns in use case modelling. Considering the misuse case is inherited from the use case, the existing work is assessed based on the following criteria: (i) the methodologies employed for modelling and specifying crosscutting security concerns and the utilization of Aspect-Oriented Modelling (AOM) features by use cases. and (ii) the misuse case and mitigation case specifications proposed in the literature to address unambiguous and formal specifications of threats and mitigations.

### Crosscutting concerns in use case modelling

The pioneer of the use case modelling approach, Ivar Jacobson, in the paper [10] has highlighted the challenge of tangling and scattering in the realization of the use case diagram. It employs the <<extend>> relationship involves defining aspects that cut across multiple use cases. It introduces the extension points as pointcuts within the base use case, and joinpoints to specify the locations where these extensions take place. The extension operation then activates advice on the extension pointcut, thereby influencing the behavior linked to the extension in the base use case. Another approach [14] proposed aspect-oriented UML use cases. It defines the crosscutting system functionality as an aspect use case and specifies it with the base use case using pointcut, joinpoint, and advice expressions. It uses the traditional use case <<extend>> relationship representing the crosscutting concerns with the extension point as a pointcut which joins at the steps in the basic flow, applying particular behavior of extension as advice. Based on this specification, the aspect use case model is further specified formally in extended AspectZ.

There are approaches in literature that provide alternative mechanisms instead of using the traditional <<extend>> relationship [15,16]. An existing approach [15] uses aspectual use cases to model crosscutting concerns as non-functional requirements (NFR) that crosscut functional use cases. The NFR is connected to the use case using stereotype <<constrain>>, and itself NFR use case is represented visually using stereotype <<NFR>>. A candidate aspect, considering a use case as a good candidate when it is related to more than one use case using the traditional extend and include relationships. Another approach [16] adapts the use cases to model and specify crosscutting behavior into distinct entities called crosscutting use cases. It represents the NFR as security, reliability, and performance as crosscutting concerns and establishes connections with regular use cases using the <<crosscuts>> relationship. It proposes the concept of a composition table to define the composition between affected use cases and their crosscutting use cases. It includes the details of affected use cases, conditions of composition, the behavior of the crosscutting use case specified as advice using operators after, before, overlap, override, and wrap, at the points in affected use cases where this behavior is applied.

Since there are approaches in literature that address aspect-oriented modelling with use cases [17–21]. An approach [17] addresses an aspect-oriented use case model that introduces an "aspects" block to consolidate new functional and

non-functional concerns, such as performance, logging, and authentication, with the core functional use cases. It incorporates aspect-oriented modelling constructs like pointcut, joinpoint, and advice within the use case model. The aspect block is linked with more than one core concern as use cases using the relationship stereotype <<crosscut>>. It consists of the pointcut on the joinpoints in the core use cases and corresponding advice as before, after, and around. Another study [18] defines an aspect-oriented method for defining and addressing security threats within the context of a use case-based functional modelling approach. It introduces a use case template to express crosscutting aspects, recognizing that threats may manifest in various sections of the same or different use cases. The approach aligns the use of pointcuts, joinpoints, and advice with the key points in a use case, such as main flow, alternate flow, precondition, and postcondition. Moreover, one of the studies [19] introduces an aspect-oriented method for defining security requirements through the utilization of aspectual use cases. This involves incorporating a pointcut to select joinpoints in the basic and alternate flows of the base use case to apply advice. This approach proves valuable in modelling functional requirements as base use cases and security requirements as aspectual use cases, enabling security personnel to identify security threats independently of functional requirements. Lee et al. [20] present a goal-driven use case. It defines the base use case, specifying the slice of system functionality used by an actor. Another use case is defined as an aspectual use case, which is also a system functionality that cuts across the base use case. It identifies the aspectual use case that affects the base use case with the defined places as joinpoints. These joinpoints describe the places in the base use case where the aspectual use case weaves its operation along with the weaving operators as insert behavior, replace behavior, or impose constraint. Another study [21] provides an aspect-oriented use case modeling approach for product lines. It considered the functional requirements as variabilities in the system and introduced a <<variability>> relationship specialization of <<aspect> relation. These variable requirements are crosscutting concerns. These variable requirements crosscut the base use case by defining the pointcut expression that signifies the joinpoint within the elements of the base use case, such as steps, operations, alternatives, extension points, and conditions. The pointcut applies the advice on the selected joinpoints using various types, such as before, after, around, and concurrent types. The variable requirements as crosscutting concerns are woven with the common requirements specified by a pointcut expression. Finally, they transformed the aspect model to a Petri net, to generate the state chart and test model through automated and extended tool support as UCEd.

There are UML profiles SecureUML [22], UMLsec [23,24], SECTET [25], ModelSec [26], and SecureMDD [27] that build UML profiles with the integration of aspect-oriented modelling to address crosscutting security concerns. SecureUML [22] proposed a Role-based Access Control (RBAC) framework to integrate crosscutting security concerns in system design. Whereas UMLSec [23,24] proposed stereotypes and constraints to specify crosscutting security concerns as confidentiality and integrity. The focus of SECTECT [25] is to model the security guidelines while modeling crosscutting security concerns in service-oriented architectures. ModelSec [26] proposed the UML profiles for class diagram, sequence diagram, and state machine diagram to model crosscutting security threats within system design. Similarly, the focus of SecureMDD [27] is on UML use case, activity diagram, and component diagram. This is observed from these studies, which highlight the importance of modeling crosscutting concerns, but they do not follow the true concepts of pointcut, joinpoint, and advice.

Despite addressing the significance of incorporating an aspect-oriented approach in use case methodology in software development, the above literature highlights several key differences.

- Few studies rely on the traditional use case <<extend>> relationship as an aspect, which causes ambiguity, particularly in identifying aspect threats or additional behavior [10,14].

- While some studies [15,16] recognize the crosscutting concerns as NFRs and specify by introducing the stereotypes as <<crosscut>>, <<aspect>>, or <<constrain>>, but they fail to fully apply the core aspect-oriented concepts such as pointcut, joinpoint, and advice.

- Literature generally lacks a comprehensive approach to model crosscutting security threats; only a few studies [17–21] specify the security threats as an aspect following pointcut, joinpoint, and advice. Aspect modeling is primarily applied for crosscutting NFR performance, reliability, security, logging, authentication, and others [15–19] or functional requirements [10,14,20,21] that crosscut with the base requirements.

- However, these approaches relied on the use case modeling method to define crosscutting security threats, rather than utilizing the more specialized misuse case model originally designed for capturing security threats. This deviation from the traditional use of use cases, typically intended for functional requirements, presents a limitation in modeling security concerns separated from the functional ones.

## Specification of crosscutting mitigations

Only one study exists that addresses the issue of tangling and scattering in the specification of mitigation cases. Dianxiang et al. [18] proposed the mitigation aspect, which identifies mitigation pointcuts and applies the mitigation advice. The proposed mitigation advice template includes the advice name, threat category, main flow, alternate flow, precondition, and postcondition. Mitigation points specify the threat points that need the mitigation logic to apply, and the mitigation advice defines the steps that mitigate the identified threat.

This study also addressed the existing mitigation templates in the literature for comprehensive coverage of the existing approaches. The purpose is to select and extend the mitigation template that can be aligned with and reflect the intended objective. Another approach presents a more comprehensive and precise mitigation scheme complementary to the security use cases to mitigate the security threat. It proposed a mitigation section, where it defines the name of the misuse case to mitigate, and its behavior using basic and alternate flows to mitigate the threats. The security use cases are complemented by the mitigation template defining the mitigation scheme [7]. Another approach provides a misuse case model for security testing. It defines the misuse cases and their mitigation in a single template, including sections such as ID, name, category, goal, actor, precondition, threat flow, post condition, and mitigation flow [28].

- From the literature, it is observed that only one study has proposed the aspect-oriented mitigation template while following the essential concepts of aspect-oriented modelling as pointcut and advice and fails to specify the joinpoints and advice types in the mitigation template.

## Unambiguous specification of misuse case

Specifying security threats unambiguously and formally is an active area. Many studies have presented the structured misuse case template to specify the security threats to reduce the ambiguity in misuse case descriptions [7,8,29]. But in general, they do not address the specification of crosscutting security threats, which still causes ambiguity in threat specification. Therefore, the basic purpose of this study is to develop an unambiguous and formalized aspect-oriented misuse case template. To achieve this objective, this study included the existing structured and formalized misuse case templates that provide an improved and structured approach with complete details. This can be further extended to specify the crosscutting security threats

El-Attar [8] presents a Structured Misuse Case Description (SMCD) to reduce the ambiguity in misuse case models by proposing a structured template and limiting the use of natural language through keywords to formally specify the security requirements. It extends the SSUCD [30] with sections such as Misuse case name, Associated Misuser, Misuser profile, Description, Extended Misuse cases, Extension points, Threatens, and Mitigates. It formalizes the structural elements as generalization, includes, and extends relationships using the formal keywords such as ABSTRACT, SPECIALIZES, IMPLEMENTS, INCLUDE, and IF keywords. The emphasis is on structured misuse cases serving as guidelines to authors in providing a comprehensive and common understanding regarding threat specification. Another approach is presented

by El-Attar [29] that defines a Formal Misuse Case Description (FMCD). It provides a formal structure for misuse case specification before transforming it into a mal-activity diagram. It restricts the use of natural language by formalizing loops and concurrency using RESUME and AFTER, and conditions using IF and AT keywords. Furthermore, it advocates initiating the basic flow with the keyword START and ending it with FINAL.

The study proposed by Mai et al. [7] reduces the ambiguity in the Restricted Misuse Case model (RMCM) by introducing a structured misuse case template and formalizing it using restriction rules. It extends the Restricted Use Case Model (RUCM) [31] with the misuse case constructs and fifteen restriction rules to specify the threats unambiguously. The proposed approach is empirically evaluated as precise, structured, and analyzable to model the security threats.

• It is evident from the above literature that the studies focused on limiting the use of natural language to reduce ambiguity in the specification of security threats. They proposed restricted keywords and rules to specify the security threats in the misuse case specification template. However, according to the objective, these approaches are lacking in specifying crosscutting security threats. Nevertheless, they are precise and unambiguous enough to be extended with AOM constructs and restriction rules to specify the crosscutting security threats.

### Research gap analysis

In general, the key limitations in existing literature are analyzed in relation to the objective of this study.

• There is a notable gap in the literature specifically addressing tangling and scattering in the context of misuse case modelling. The existing approaches typically rely on use case diagrams and specification templates to define crosscutting security threats, rather than utilizing the more specialized misuse case model designed for capturing security threats. This reliance on traditional use cases typically intended for functional requirements presents a limitation in modelling security concerns separately from functional ones.

• The existing aspect-oriented use cases cannot specify the introduction, advice (before, after, and around), and weaving to model crosscutting security threats and their mitigations.

• While several misuse case and mitigation specification templates exist to best address the threats and mitigation precisely and unambiguously. But to the best of current knowledge, no one has specified crosscutting security threats and their mitigation using such templates.

In the context of this study, there is a need to propose a comprehensive and unambiguous aspect-oriented approach that specifies the crosscutting security threats and their mitigation using misuse case and mitigation case templates. Which follows the essential aspect-oriented concepts such as pointcut, joinpoint, advice, introduction, and weaving. For this purpose, among the literature, one of the most comprehensive and precise approaches is RMCM [7]. This study has adopted and extended the RMCM template and restriction rules to specify aspect-oriented misuse cases and aspect-oriented mitigation cases to reduce ambiguity and effectively specify the crosscutting security threats and their mitigations.

### Overview of the proposed approach

This section provides an overview of the proposed AFMUC approach for specifying the crosscutting threats and their mitigations as shown in Fig 1. The aspect-oriented constructs are integrated with the misuse case model to specify and analyze the crosscutting security threats and their mitigations.

In Fig 1. three roles are presented as *Approach Provider*, *Application Modeler,* and *Security Expert*. The approach provider provides aspect-oriented guidelines for modeling the AFMUC approach, and the security expert models the threats and mitigations in the misuse cases using the guidelines and weaves the aspects in the misuse case and mitigation case

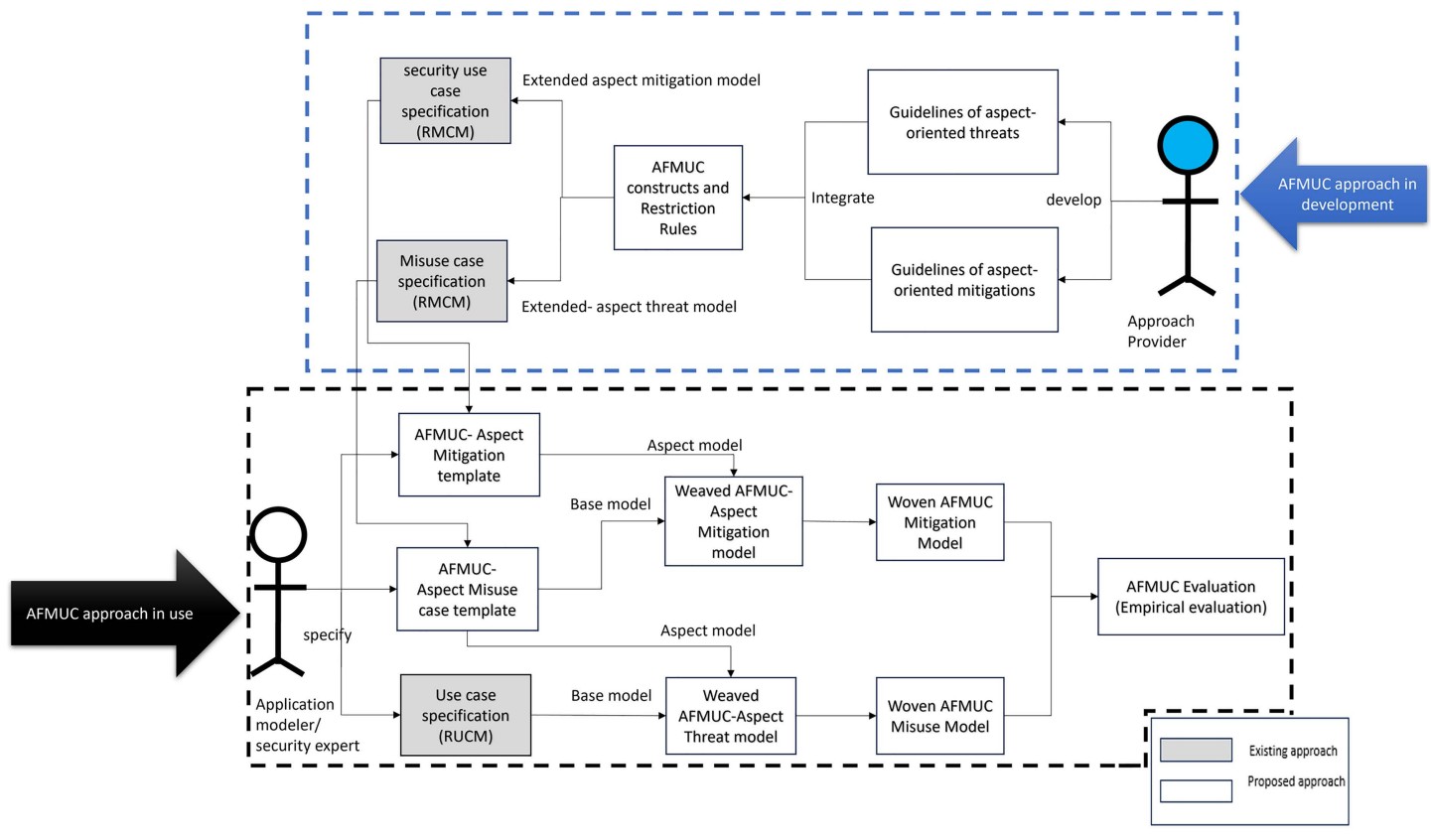

**Fig 1. Overview of proposed AFMUC approach.**

specifications resulting in woven AFMUC models. As presented in Fig 1. in the AFMUC approach development phase, the approach provider utilizes the existing RMCM [7] and its security template to extend with the aspect-oriented constructs and restriction rules, which results in the aspect-oriented formalized misuse cases AFMUC template. It further provides guidelines to specify the aspect threats and aspect mitigations for AFMUC modelling. Then, in AFMUC approach in use phase, the application modeler specifies the functional requirements using the Restricted Use Case Modelling (RUCM) [31]. Based on the specified functional requirements, the security expert specifies and analyzes the aspect threats in AFMUC misuse case template for each of the identified functionalities. Further, the security expert specifies the AFMUC mitigation cases to mitigate the identified aspect threats. Moreover, the security expert uses AFMUC guidelines and develops a woven AFMUC aspect threat and mitigation model. These woven AFMUC misuse and mitigation models are evaluated to validate the ambiguity of the proposed AFMUC approach.

The following sections discuss the proposed approach in detail.

## Requirements specification

The artifacts used by the AFMUC are the use case specification, misuse case specification, and security requirements specification. The functional requirements are specified using the RUCM template proposed by Yue et al. [31]. In proposed context, the essential required sections of this template are, *use case name*, *Dependency*, *Precondition*, B*asic Flow*, and Post*condition*. Based on the specified use case, the threats and mitigations are specified using the misuse cases and security use cases respectively, by adopting the RMCM template [7]. These templates are selected because

they are renowned, actively used by the researchers, and evaluated as precise, structured, and practical to analyze the functional and security requirements [32,33].

## AFMUC constructs

The basic constructs of AOM are *pointcut*, *joinpoint, advice, introduction, and weaving* [34]. The aspects for threats and mitigations are defined and then woven into the misuse case and mitigation case description templates, respectively. This section explains AOM constructs used in the proposed AFMUC (misuse case and mitigation case) approach.

**Aspect.** Aspect is defined as any concern that cuts across multiple components in the base model [34]. In AFMUC, the language construct *aspect* is defined as an *aspect misuse case* that represents a misuse case that cuts across multiple use cases of the software application. It is known as an *aspect threat*, for example, "unauthorized access". Also, an *aspect mitigation* represents the mitigation case to prevent multiple misuse cases (i.e., threats), for example, "monitor network". Each aspect defines its properties as joinpoints, pointcuts, advice, and introduction.

**Pointcut.** Pointcut is an expression that selects one or more joinpoints where advice could be applied [34]. In the context of aspect threat, pointcut expression defines the name of a use case where the aspect crosscut is based on the condition to select the attack surfaces in the use case elements as a precondition, basic flow, or postcondition. For the aspect mitigation, the pointcut expression specifies the name of the misuse case along with the condition to select the mitigation surfaces from the misuse case elements as a precondition, basic threat flow, or postcondition.

**Joinpoint.** Joinpoint(s) is the location in the base model where the advice is applied [34]. In the context of aspect threats, the joinpoints are the sequence numbers in the use case elements as attack surfaces satisfying the condition specified in the pointcut expression. Similarly, for the aspect mitigation, the joinpoints are the mitigation surfaces that exist in the misuse case elements as sequence numbers satisfying the pointcut condition.

**Advice.** Advice is the behavior that is applied to the selected joinpoints. It has three types: (i) *before advice*: this advice is executed before the selected joinpoints, (ii) *around advice*: this advice replaces the joinpoints, and (iii) *after advice*: this advice is executed after the selected joinpoints [34]. In the context of aspect threat, the advice refers to the basic threat flow that is applied at selected joinpoints (i.e., attack surfaces) to execute the threat. When this basic threat flow applies before the basic flow of a use case (precondition), then it is referred to as *before advice*. The basic threat flow applied at a use case basic flow is known as *around advice*. The basic threat flow applies after the use case basic flow (postcondition) is called *after advice*. In the context of aspect mitigation, the advice is the basic mitigation flow, which is applied at the selected joinpoints (i.e., mitigation surfaces) to prevent the threat. *Before advice* is applied at precondition, before the threat execution (i.e., basic threat flow). *Around advice* replaces the basic threat flow, whereas *after advice* is applied at postcondition upon completion of basic threat flow.

**Introduction.** Introduction is the addition of a new behavior that never existed before in the base model [34]. In AFMUC context, the introduction is any new aspect threat added to the application. This can occur by replacing the existing use case or associating it with the existing use case through include or extend relationship. Concerning aspect mitigation, the introduction is not applicable because new mitigations cannot be introduced into the application arbitrarily. Mitigations are designed based on identified threats. If a threat exists, the corresponding mitigation can be implemented depending on the nature of the threat. Therefore, no new mitigation can be introduced into the application if no threat has been analyzed.

**Weaving.** Weaving is the process of systematically integrating the aspect model into a base model at selected joinpoints, which are specified by pointcut expressions, to apply the corresponding advice behavior [34]. In the AFMUC context, the aspect model is an aspect misuse case that integrates with the base use case model while specifying the selected joinpoints (attack surfaces in use cases), defined by a pointcut expression to apply the threat behavior (basic threat flow) as advice. In a mitigation case, the aspect model is "aspect mitigation" that integrates with the base misuse case model and applies the mitigation behavior as basic mitigation flow at the selected joinpoints (mitigation surfaces in misuse cases) specified by pointcut.

## Aspect-oriented formalized misuse cases to specify aspect threats

Table 1 shows the proposed AFMUC – aspect misuse case template for the specification of aspect threats in a precise and structured form. It extended the RMCM template [7] with aspect-oriented modeling constructs such as pointcut, joinpoint, advice, and introduction.

The RMCM [7] approach includes elements such as misuse case name, *brief description, primary actor, precondition, basic threat flow, alternate threat flow, alternate flow, and dependency*. In this study, all elements from the RMCM have been adopted, except for the *misuse case name* and threat section; these two elements are modified and renamed as *AFMUC name* and *Threat Pointcut.* The *AFMUC name* represents the crosscutting threat as an aspect misuse case name. This differentiation is made to distinguish the regular misuse case name as an aspect threat name to indicate it as an aspect. The *threat pointcut* section replaces the threat section. Previously, the threat section only specified the name of the use cases it threatens, causing ambiguity as it fails to provide comprehensive details on how a threat is cut across use cases. The proposed *Threat Pointcut* section addresses this issue by specifying the crosscutting threats using AOM constructs such as pointcut, joinpoint, advice, and introduction. Additionally, *AFMUC ID is introduced* to specify the unique

**Table 1. AFMUC- Aspect Misuse Case Template.**

| *AFMUC ID* | *The unique identifier of the aspect misuse case.* | |
|---|---|---|
| *AFMUC Name* | *Specify the name of aspect threat that cuts across use cases.* | |
| Brief Description | Briefly describe the aspect misuse case. | |
| Precondition | Specify the condition to satisfy before execution of threat flow | |
| Primary Actor | The malicious user who initiates the aspect misuse case | |
| Secondary actors | The actors who interact with the system to accomplish the misuse case. | |
| Dependency | Include and extend relationship to other mis(use) cases. | |
| Generalization | Generalization relationship to other misuse cases. | |
| Assets | The assets (potentially) impacted by this threat. | |
| *Threat category* | *The main category of threat is compliance with standards.* | |
| *Threat pointcut* | *Pointcut* | *Specify the expression to select the joinpoints from the use case elements as precondition, postcondition, and basic flow* |
| | *Joinpoint* | *Specify the attack surface as sequence numbers from the precondition, postcondition, and basic flow, where the aspect threat pointcut* |
| | *Advice* | *Apply advice with its type as the Before, After, or Around advice* |
| | *Introduction* | *Add threatening behavior to existing use cases without changing its own functionality.* |
| Basic Threat Flow | Specify the main sequence of action steps that harm the system | |
| | Steps(numbered) | Flow of events |
| | Post condition | The resulting unwanted condition and the asset(s) impacted after the threat flow executes. |
| Specific /Bounded /Global Alternative Threat Flow | A specific alternative sequence of actions carried out by a misuser to harm the system. | |
| | RFS | A reference flow step number where flow branches from. |
| | Steps (numbered) | Flow of events |
| | Post condition | The resulting unwanted condition and the asset(s) impacted after the threat flow. executes. |
| Specific /Bounded /Global Alternative Flow | A specific alternative sequence of actions that does not result in any harm to the system. | |
| | RFS | A reference flow step number where flow branches from. |
| | Steps (numbered) | Flow of events |
| | Postcondition | The resulting condition after the alternative flow executes. |
| Mitigation scheme | Refers to the name of the mitigation scheme, specified using mitigation template, to mitigate this misuse case. This complements security use case(s). | |

identifier of the threat, and the *Threat Category* sections represent its compliance with the OWASP [35] and ISO/IEC 27001:2022 Information Security Management [36] Standards. The complete details of the RMCM template can be found at [7].

## Aspect-oriented formalized mitigations to specify aspect mitigation

AFMUC also provides the aspect mitigation template presented in Table 2. The RMCM [7] security use case specification template is adopted, and aspect-oriented constructs are integrated to represent the crosscutting mitigation of the threats. The elements, such as *security use case name*, *mitigation*, and *basic flow* of security use case specification, are replaced with the proposed AFMUC constructs as *Aspect Mitigation Name*, *Mitigate Pointcut, Basic Mitigation Flow,* and *Alternate Mitigation Flow*. The *Aspect Mitigation Name* represents the name of mitigation that cuts across multiple threats or misuse cases. *Basic Mitigation Flow* represents the behavior used to mitigate threat, *Alternate Mitigation Flow* specifies the alternate behavior used to mitigate threat in reference to basic mitigation flow. A field of *Aspect Mitigation ID* is also introduced to assign a unique identifier to the aspect mitigation. This approach is used to provide relevant names and to differentiate them from other templates. Lastly, the *Mitigate Pointcut* represents the actual specification of AOM constructs such as pointcut, joinpoint, and advice.

## Aspect-oriented formalized restriction rules

The aspect-oriented restriction rules are designed to formally and unambiguously specify the crosscutting security threats and their mitigation. These rules apply to aspect threats and aspect mitigations. These proposed rules are extensions of the RMCM restriction rules [7], as they have already been assessed as unambiguous, precise, and practically applicable. Altogether, the AFMUC template and restriction rules are designed to ensure that the aspect threat and aspect mitigation are unambiguous, structured, and formalized by restricting the use of natural language in specifying the crosscutting security threats and their mitigation.

Table 3 represents sixteen restriction rules (R1-R16) for specification of aspect threats and their aspect mitigation using pointcut, join point, advice, and introduction. The rules (R1-R5) are used to formally specify the elements (subsets) of the

**Table 2. AFMUC- Aspect Mitigation Template.**

| Aspect Mitigation ID | Specify the unique ID of aspect mitigation | |
|---|---|---|
| *Aspect Mitigation Name* | *Specify the name of aspect mitigation* | |
| Brief Description | Summarizes the mitigation case with a short paragraph | |
| Dependency | Included in use cases | |
| *Mitigate pointcut* | *Pointcut* | *Specify the expression to select the joinpoints from the misuse case elements as precondition, postcondition, and basic threat flow* |
| | *Join point* | *Specify the mitigation surface as sequence numbers from the precondition, postcondition, and basic threat flow where the mitigation threat pointcut* |
| | *Advice* | *Apply advice with its type as the Before, After, or Around advice* |
| Precondition | Specify the condition to satisfy before execution of mitigation | |
| *Basic Mitigation Flow* | *Specify the main sequence of action steps that protect the system* | |
| *Alternate Mitigating Flow* | *Specify the alternate sequence of actions that protect the system* | |
| Postcondition | The resulting condition after the execution of mitigation | |

**Table 3. AFMUC- Restriction Rules.**

| # | Rules | Explanation |
|---|---|---|
| R 1. | BF | Represents the subset as <u>Basic Flow</u> in the use case specification. |
| R 2. | PRECONDITION | Represents the subset as <u>Precondition</u>. |
| R 3. | POSTCONDITION | Represents the subset as <u>Post condition</u>. |
| R 4. | BTF | Represents the subset as <u>Basic Threat Flow.</u> |
| R 5. | BMF | Represents the subset as <u>Basic Mitigation Flow.</u> |
| R 6. | POINTCUT<name> SUBSET <selection constraint> | Represents the pointcut expression to select the joinpoints. |
| R 7. | JOINPOINT <sequence no> | Represents the joinpoints as sequence numbers which satisfy the pointcut expression. |
| R 8. | ADVICE<type> POINTCUT <sequence no> EXECUTE <flow> | Represents the aspect behavior applied on the selected joinpoints defined by the pointcut. |
| R 9. | BEFORE | Represents the type of advice to execute the aspect behavior at the precondition. |
| R 10. | AFTER | Represents the type of advice to execute the aspect behavior at the postcondition. |
| R 11. | AROUND | Represents the type of advice to execute the aspect behavior at the R1, R4. |
| R 12. | && | Represents the AND logical operator used to must combine two or more conditions, joinpoints, pointcuts or use cases. |
| R 13. | \|\| | Represents the OR logical operator used to combine any of two or more conditions, joinpoints, pointcuts or use cases. |
| R 14. | ALL | Represents ALL the conditions, joinpoints or pointcuts. |
| R 15. | INTRODUCTION <name> ASSOCIATE TO <use case> AS <dependency> | Represents the aspect threat as a new functionality and associate with the existing use case as INCLUDE USE CASE or EXTENDED BY USE CASE. |
| R 16. | INTRODUCTION <name> REPLACE <use case> | Represents the aspect threat as a new functionality and replaces the existing use case. |

description templates (*Basic Flow, Precondition, Post Condition*, *Basic Threat Flow, Basic Mitigation Flow*) in the relevant templates. The rules (R6-R16) are used to formally specify the AOM constructs as *Pointcut, Joinpoint, Advice,* and *Introduction*.

## Formalizing the aspect threats using restriction rules

The AFMUC restriction rules are mentioned in the *Threat Pointcut* section of the template (Table 1), that are used to define the aspect misuse case model To define the aspect misuse case (i.e., aspect threat), the rules R1-R4 and R6-R16 except the R5 are mandatory to complete the expression with essential constructs, i.e., pointcut, joinpoint, advice, and introduction. The R5 is not applicable here because it specifies the basic mitigation flow, mostly useful in specification of aspect mitigation model.

Referring to rule R6, the POINTCUT expression **(POINTCUT<name> SUBSET <selection constraint>)** selects the base use case in **<name>** parameter. The SUBSET reflects use case element specified as **BF** (R1), **PRECONDITION** (R2), or **POSTCONDITION (**R3). The **<selection constraint>** is a condition to select the attack surfaces (joinpoints) from the SUBSET. **JOINPOINT** (R7) is an expression **(JOINPOINT <sequence no>)** used to specify the attack surfaces with the sequence numbers that satisfy the selection constraint defined in R6. R6 and R7 collectively form a complete pointcut expression that selects the joinpoints where advice can be applied. To apply advice, all relevant pointcuts and joinpoints must be defined beforehand. Advice is defined as an expression **(ADVICE <type> POINTCUT <sequence no> EXECUTE <flow>)** used to specify the threat behavior. It is applied to the selected joinpoints identified by the pointcut expression. It consists of three parameters where the parameter **<type>** specifies the advice type: **BEFORE** (R9), **AFTER** (R10), or **AROUND** (R11). **BEFORE** Advice indicates that the SUBSET in POINTCUT expression serves as a **PRECONDITION** (R2) of the base use case where JOINPOINT exists, similarly the **AFTER** Advice reflects the SUBSET

as **POSTCONDITION** (R3) of the base use case, and the **AROUND** advice applied at JOINPOINT exists at **BF** (R1) of the base use case. The **ADVICE** executes the threat behavior as BTF (R4), representing "Basic Threat Flow" in the **<flow> parameter.** This behavior is applied to POINTCUT expressions with their numbers given in parameter **<sequences no>**. Alternatively, the keyword **ALL** (R14) can be used if **ADVICE** applies to all specified POINTCUT expressions.

Regarding rule R15, the INTRODUCTION expression is defined to add a new threat to the existing base use case, showing it is a legitimate functionality. The expression **(INTRODUCTION <name> ASSOCIATE TO <use case> AS <dependency>)** specifies the threat name specified in the **<name>** parameter, which is introduced in the application by associating it with the existing base use case in parameter **<use case>**. The association relationship between the newly added threat and the existing base use case specified in the **<dependency>** parameter. This association may be both **INCLUDE USE CASE** or **EXTENDED BY USE CASE**. Either Rule R13 (||) or R14 (&&) is used to associate the threat with multiple use cases.

The second form of INTRODUCTION expression (R16), **INTRODUCTION <name> REPLACE <use case>,** is used to replace the base use case with the threat. The **<name>** parameter defines the name of the threat, while the **<use case>** parameter is used to define the name of base use cases to be replaced. Rule R 13 (||) or R14 (&&) may be used to select multiple use cases to replace them with a threat.

### Formalizing the aspect mitigations using restriction rules

The restriction rules (R2-R14) in Table 3 are applicable to specify the aspect mitigation using AFMUC aspect mitigation template (Table 2). While the purpose of these rules remains consistent, their interpretation changes in this context. This is because the base model refers to a misuse case, while the aspect model is an aspect mitigation case. For example, the rule R6 **(POINTCUT<name> SUBSET <selection constraint>)** represents the name of the base misuse case as a POINTCUT in **<name>** parameter. The parameter **<selection constraint>** is used to define the condition to select the joinpoints within misuse case **SUBSET,** such as **PRECONDITION** (R2), **POSTCONDITION** (R3), or **BTF** (R4). The Joinpoints **(JOINPOINT <sequence no>)** select the mitigation surfaces within the **SUBSET** that satisfy the selection constraint. The ADVICE expression **(ADVICE <type> POINTCUT <sequence no> EXECUTE <flow>)** is used to execute the mitigation behavior as **BMF** (R5), on the selected joinpoints. The **<type>** parameter in the **ADVICE** expression determines advice type. **BEFORE** (R9) advice applies when the SUBSET in the POINTCUT expression is **PRECONDITION** (R2), the purpose is to prevent the threat from occurring. **AFTER** (R10) advice is applied when SUBSET is **POSTCONDITION** (R3). It is used to apply post-mitigations after the threat has harmed the application. And lastly, **AROUND** (R11) advice is applied at SUBSET as **BTF** (R4) and is used to mitigate threat by controlling its execution.

Another difference is that R1, R15, and R16 are not applicable in aspect mitigation. Specifically, R1 is relevant for selecting attack surfaces as joinpoints from the base use case model and is utilized in aspect threat specification, as discussed in section "Aspect-Oriented Formalized Misuse Cases to Specify Aspect Threats". On the other hand, rules R15 and R16 are not applicable because the aspect construct *introduction* is intended to add a new behavior to the application without altering the existing behavior. Accordingly, mitigation inherently involves responding to an applied threat. Therefore, it requires the existence of a threat and cannot be introduced independently.

### Example case study

This section demonstrates the applicability of the proposed AFMUC approach on an excerpt from case study, EU-rent car [37]. EU-Rent car is a web application providing car rental services across various towns in multiple countries via the internet, featuring a thousand branches. The EU-rent car application provides several features like customer management, car reservations and others. The customer can register, reserve cars, get membership, pay online, get offers, and return a car to any branch in town. EU-Rent, a car web application, allows customers to enter their details for registration and then make online payments to get membership and reservations. Ensuring security and data protection is a major challenge as

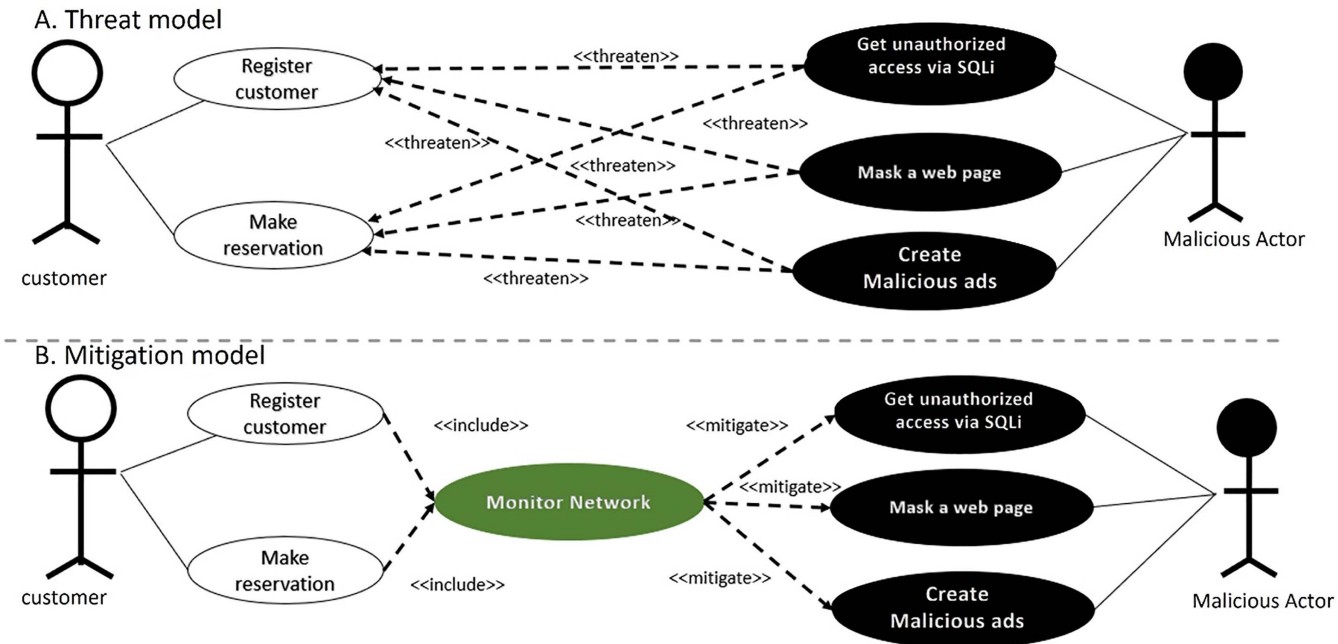

**Fig 2. Excerpt of misuse case diagram - EU rent a car.**

this feature cuts across multiple functionalities in the application. Moreover, it processes the customer's sensitive data and shares it on the network while registering, reserving cars, getting a membership, and others.

The excerpt of the EU-rent car application in Fig 2 (A, B) represents the misuse case diagram as a threat model and mitigation model. Two use cases, *register customer* and *make reservation,* are selected to demonstrate the proposed AFMUC approach. Their details are specified using the RUCM [31] template in Fig 3. The customer can register their account by providing details such as name, address, date of birth, and driving license. Another functionality of an application is to make a reservation using the assigned ID. After registering, the customer provides the details of car and pickup/drop off branches. Finally, the reservation is confirmed upon credit card payment. It analyzes the application process and stores the customer's sensitive data. The application is vulnerable to multiple threats, such as unauthorized access, data modification, man-in-the-middle attack, denial of service, defacement, and phishing, which crosscut these use cases. For aspect threats modelling, *unauthorized access via SQLi, mask a web page,* and *create malicious ads* are considered. The specification of threats using RMCM is given in Fig 4. These threats attack the given use cases while identifying the attack surfaces. To prevent these threats, there are many mitigation strategies. The excerpt of mitigation is provided as "*monitor network*" which cuts across identified threats by implementing the intrusion prevention system (IPS). To monitor the network, IPS prevent these threats when it finds suspicious activity in the network these. The mitigation specification is given using the RMCM security use case template in Fig 5.

## Application of proposed approach

Fig 6 (A, B) shows that the aspect threat "*get unauthorized access via SQLi*" cuts across two use cases named *register customer* and *make reservation*. In Fig. 6 (B), the specification of aspect threat shows that *Get Unauthorized Access* via *SQLi* is an *AFMUC name*. In the "*Threat Category*" section, the type of threat as an injection attack is specified in compliance with the OWASP [35]. The AFMUC element "*Threat Pointcut*" specifies a crosscutting threat aspect including the pointcuts, joinpoints, and advice. In this section, the proposed restriction rules are applied as: the keyword *POINTCUT*

| Use case name | Register customer |
|---|---|
| Dependency | None |
| Precondition | 1. The customer initiates the registration URL.<br>2. The user is not an existing customer.<br>3. The customer has network connection. |
| Basic flow | 1. The customer enters the name to the system.<br>2. The customer enters the address to the system.<br>3. The customer set the date of birth on calendar.<br>4. The customer enters the driving license number to the system.<br>5. The customer selects the date of issue of driving license from calendar.<br>6. The customer selects the date of expiration of driving license from calendar.<br>7. The system VALIDATES THAT the information is correct.<br>8. The system shows welcome message to the customer. |
| Post condition | 1. The customer is registered.<br>2. The customer DATA is stored in the database with unique identifier. |
| Use case name | Make a Reservation |
| Dependency | INCLUDE USE CASE offer points payment<br>INCLUDE USE CASE offer special advantages |
| Precondition | 1. The customer initiates the make reservation URL.<br>2. The customer is registered in the system.<br>3. The customer has network connection. |
| Basic flow | 1. The customer enters customer ID in the system.<br>2. The system VALIDATES THAT the customer is not backlisted.<br>3. The customer selects the desired period of reservation to the system.<br>4. The system VALIDATES THAT the desired period of reservation is correct.<br>5. The customer selects the pickup branch to the system.<br>6. The customer selects the drop off branch to the system.<br>7. The customer selects the visiting countries to the system.<br>8. The customer selects the car group to the system.<br>9. The customer selects the desired car model to the system.<br>10. IF the customer enters the credit card number, THEN the rental is guaranteed.<br>11. IF the customer is a member of the loyalty incentive scheme, THEN initiate INCLUDED USE CASE offer points payment ELSE initiate INCLUDED USE CASE offer special advantages.<br>12. The customer confirms the rental.<br>13. The system creates a rental agreement. |
| Post condition | 1. Rental agreement is generated.<br>2. The customer reservation DATA is stored in the database.<br>3. The customer downloads the agreement |

**Fig 3. Excerpt of EU rent a car (Use Case).**

(R6) is used to explicitly define the name of the base use cases as *register customer* and *make reservation*, along with the SUBSET as *BF* (R1) having the selection constraint as "customer enters data". According to the aspect threat, the attacker can get unauthorized access to the application when the input fields are vulnerable to entering a malicious SQLi script. Next, the rule (R7) defines the keyword *JOINPOINT* along with the sequence steps selections of the attack surfaces from BF, which satisfies the selection constraint "customer enters data" *(highlighted in red color and annotated as joinpoints -Fig 6 (A))*. Therefore, the joinpoint expression for *register customer*, is *JOINPOINTS 1 && 2 && 4,* and for the use case named *make reservation*, *JOINPOINT 1 && 10*. Following the R6 and R7, all the pointcut expressions are

| Misuse case name | Get unauthorized access via SQLi |
|---|---|
| Brief Description | A malicious actor attempts to gain unauthorized access to a system or application by exploiting a SQL Injection (SQLi) vulnerability. |
| Threats | Register customer, make reservation |
| Precondition | System has at least one registered user. The system's network security is inadequate for detecting spoofed IP addresses The webpage has web form. |
| Basic Threat Flow | 1. The MALICIOUS user PROVIDES SQLI VALUES IN input fields of the url. 2. The MALICIOUS user BYPASS the validation REQUEST TO server program. 3. The system executes the query provided in the url. 4. The system evaluates the query in the database. 5. The system VALIDATES THAT the query is successful. 6. The system SENDS the welcome message TO the MALICIOUS user. |
| Post condition | The MALICIOUS user accesses the customer DATA without authorization |
| Misuse case name | Mask a web page |
| Brief description | Obscure or hide the content of a web page from the user's view. |
| Threats | Register customer, Make reservation |
| Precondition | The system follows insecure coding practices. The system has misconfigurations. The system's network security is inadequate for detecting spoofed IP addresses |
| Basic Threat Flow | 1. The MALICIOUS user crafts a payload with code snippet. 2. The MALICIOUS user EXPOLITES the code injection vulnerability of server. 3. The MALICIOUS user GETS code FROM the code repository. 4. The MALICIOUS user PROVIDES REC VALUES IN the server code. 5. the MALICIOUS user SENDS a MALICIOUS payload TO the server program. 6. The MALICIOUS code BYPASS the secure code validation REQUEST TO server. 7. The system VALIDATES THAT the payload is valid. 8. The MALCIOUS user modify the server code with desired content. |
| Post condition | The MALICIOUS user successfully defaces the target website. |
| Misuse case name | Create malicious ads |
| Threats | Register customer, Make reservation |
| Precondition | The system follows insecure coding practices. The system has misconfigurations The system network security is inadequate to detect spoofed IP addresses |
| Basic Threat Flow | 1. The MALICIOUS user crafts a MALCIOUS ad with code snippet. 2. The MALICIOUS user EXPOLITES the code injection vulnerability of server. 3. The MALICIOUS user GETS code FROM the code repository. 4. The MALICIOUS user PROVIDES REC VALUES IN the server code. 5. the MALICIOUS user SENDS a MALICIOUS ad TO the server program. 6. The MALICIOUS popup BYPASS the secure code validation REQUEST TO server. 7. The system VALIDATES THAT the MALICIOUS ad is valid. 8. The system displays a MALICIOUS ad in the url. |
| Post condition | The MALICIOUS user successfully creates a malicious ad in the system. |

**Fig 4. Excerpt of EU rent a car (Misuse Case).**

defined in sequences. Lastly, the advice is applied using *"ADVICE AROUND POINTCUT ALL EXECUTE BTF"*, which implements the threat behavior on all pointcut expressions.

The overall *Threat Pointcut* section depicts that the aspect misuse case *get unauthorized access via sqli* POINTCUTS at the use cases *register customer* and *make reservation* where the constraint is customer enters data, given in the use

| Security use case name | Monitor network |
|---|---|
| Brief Description | Involves analysing network traffic patterns, protocols, and data flows to identify abnormal behaviour, spoofed IPS and detect security threats |
| Mitigate | Get unauthorized access via SQLi, mask a page, create malicious ad |
| Precondition | The Intrusion Prevention (IPS) installed within the network infrastructure. The network is operational and monitoring traffic. |
| Basic Flow | 1. The IPS continuously monitors incoming and outgoing network traffic, analysing packet headers and payload content. 2. The system utilizes signature-based detection mechanisms within the IPS to match patterns of known traffic patterns. 3. IPS detect deviations from normal network behaviour. 4. IPS decrypt and inspect encrypted traffic. 5. IPS detect Address Resolution Protocol spoofing or MAC address anomalies within the network. 6. IPS generates real-time alerts or triggers predefined actions. 7. IPS isolate the affected segments of the network. 8. IPS block the malicious traffic. |
| Alternate Flow | RFS 2 1. The IPS mistakenly identifies legitimate traffic as spoofed. 2. The IPS drops legitimate packets. RFS 5 1. The IPS does not identify the spoofed IP. 2. The spoofed IP packets are not flagged and continue to the network. 3. The network is compromised. |
| Post condition | Spoofed IP addresses are detected and mitigated. The network remains secure from the intrusion attempt. |

**Fig 5. Excerpt of mitigation specification - EU rent a car.**

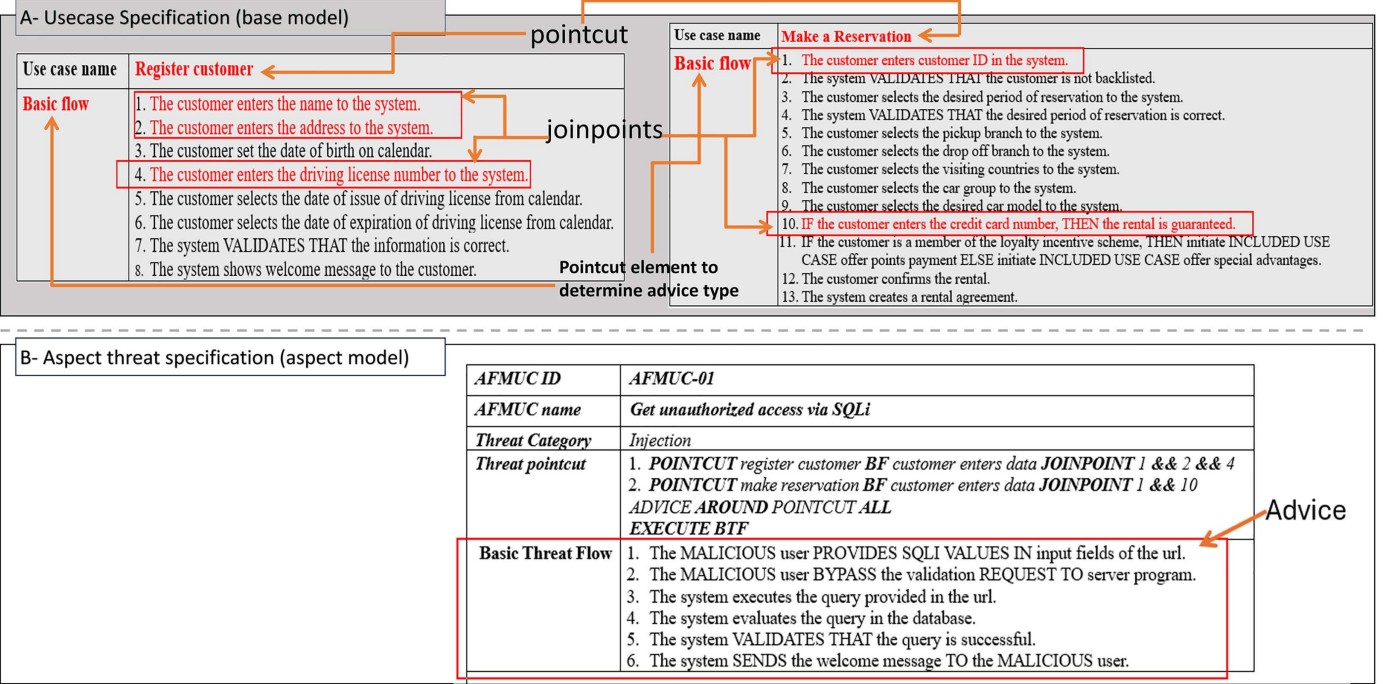

**Fig 6. Excerpt of aspect threat - pointcut joinpoint and advice.**

case BF against which the JOINPOINT are selected as the sequence steps (1&& 2&& 4, 1&&10) in BF respectively, where the ADVICE apply the basic threat flow as BTF using advice type as AROUND in ALL specified pointcuts using keyword EXECUTE.

Fig 7 (B) shows the application of R16 for the aspect threat *mask a web page* that threatens the use cases named *register customer* or *make reservation* using the keyword INTRODUCTION. It replaces the flow of the entire use case with malicious behavior by defacing it. Fig 7 (A) shows the red highlighted and annotated use cases. It shows that complete use case specification is replaced with the threat specification. This means that when the customer visits the registration or reservation page, they are redirected to a fake page that appears legitimate to the customer. This deception leads the customer to provide sensitive information to the attacker.

Similarly, Fig 8 (A, B) shows the application of R15, where the aspect threat C*reate malicious ad*, a type of phishing attack that threatens the use cases *register customer* or *make reservation*. In the *Threat Pointcut* section (Fig 8B), the keyword INTRODUCTION is used to define the threat name as *Create mal ad* and ASSOCIATE TO the use cases as *register customer* or *make reservation* with the relationship as INCLUDED USE CASE, the change is assumed to be reflected in the dependency section of the use case highlighted red and annotated dependency in Fig 8A. It means that the attacker modifies the *register customer* code (vulnerable to attack) with the malicious ad code and displays it on the registration page as a legitimate ad to the customer. And following the semantics of the include relationship, it mandates the customer to click on the ad to complete the registration. The specified threats "*get unauthorized access via SQLi*", "*mask a web page*", and "*create malicious ads*" are woven into the AFMUC misuse case template given in Fig 9.

In Fig 10B, the threats are mitigated by the mitigation case named *monitor network. It* crosscuts three threats specified in Fig 10A. The *Mitigation Pointcut* element specifies the three pointcut expressions (R6) for the threats

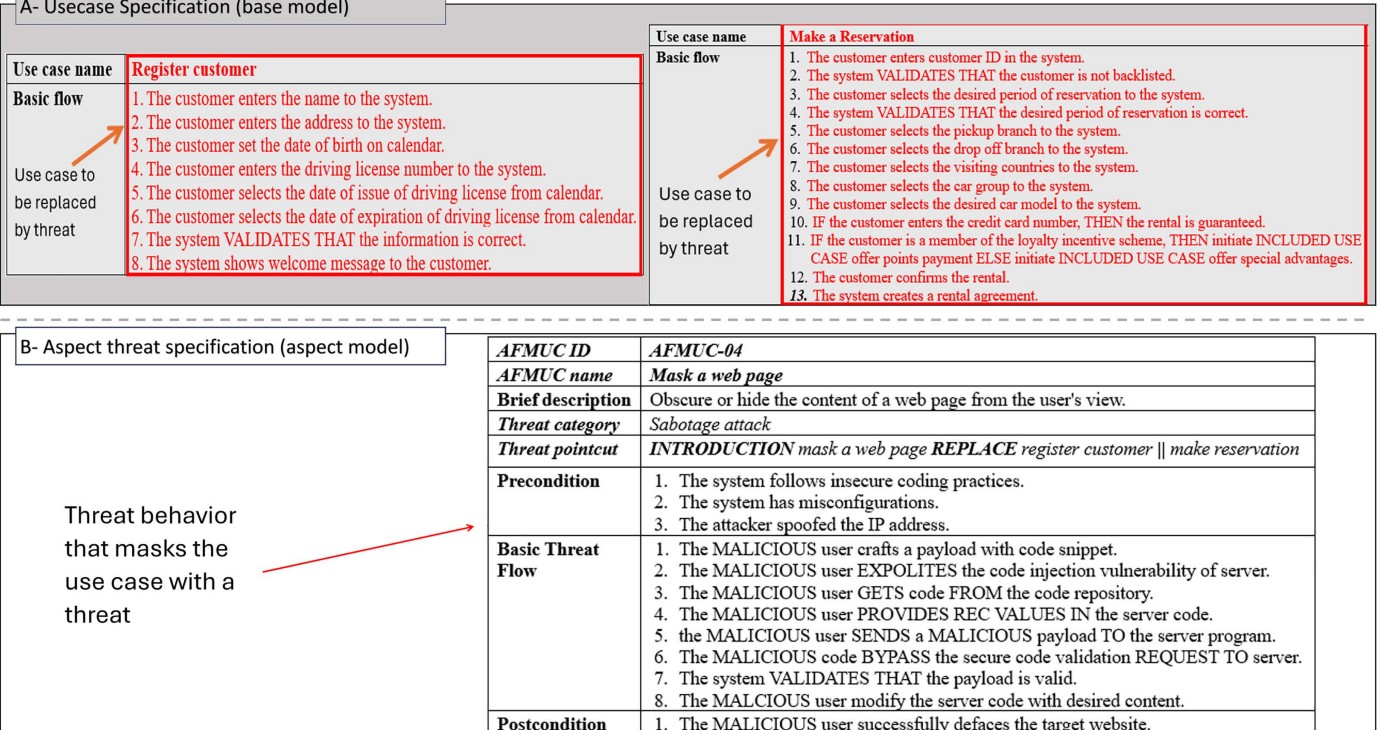

**Fig 7. Excerpt of aspect threat using introduction (R16).**

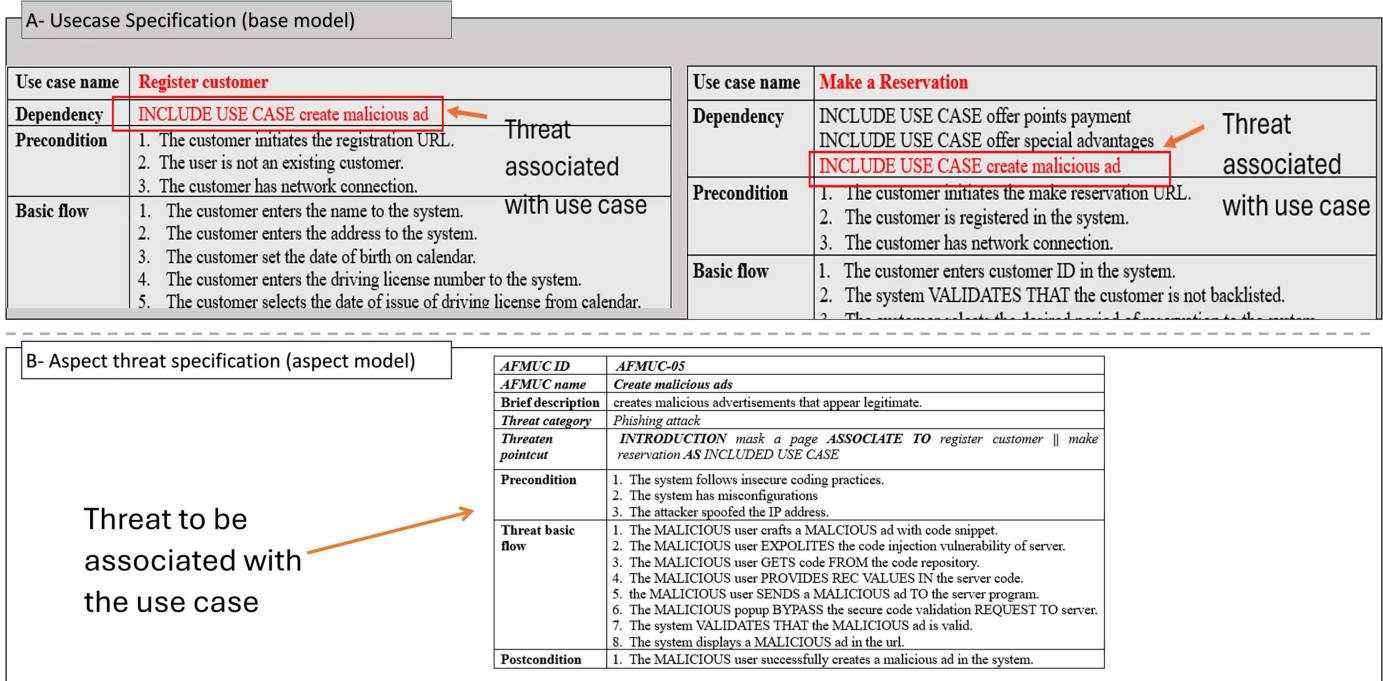

**Fig 8. Excerpt of aspect threat using introduction (R15).**

named *get unauthorized access via SQLi*, *mask a web page,* and *create malicious ads*. The selection constraint for choosing the joinpoint in all these pointcuts is defined as "spoofed IP address". Satisfying this selection constraint, the JOINPOINTS, or mitigation surfaces, for these threats are specified as precondition number four for "*get unauthorized access via SQLi*" and precondition number three for both "*mask a web page*" and "*create malicious ads*". The mitigation surfaces are highlighted in green color and annotated as joinpoints (Fig 10A). The mitigation aspect applies the basic mitigation behavior (BMF) as BEFORE advice on ALL three POINCUT expressions. This means that the spoofed *IP* address is mitigated by the intrusion prevention system via network monitoring before the execution of these threats and blocks the malicious network traffic. Fig 11 shows the complete woven model of AFMUC aspect mitigation template.

## Empirical evaluation

This section empirically evaluates the proposed AFMUC approach in an industrial case study. A controlled experiment is conducted to validate the AFMUC unambiguity in specifying crosscutting security threats and mitigations.

### The mobile finance application

The Mobile Finance Application (MFA) case study is obtained from the industry. To maintain the confidentiality agreement, the organization name is anonymized as "Company X". The acquired software requirements specification for MFA includes its functional and security requirements. It is important to note that this application is neither a practical industrial project nor a student project, but rather a comprehensive application specification designed by industrial security experts and faculty of the Information Security Department based on the provided requirements.

The objective of MFA application is to provide easy access to financial services to customers via mobile phones or a web portal, offering identical features across both platforms. This study focuses on the web-based services of the MFA

| AFMUC ID | AFMUC-01 |
|---|---|
| AFMUC name | *Get unauthorized access via SQLi* |
| Brief Description | A malicious actor attempts to gain unauthorized access to a system or application by exploiting a SQL Injection (SQLi) vulnerability. |
| Primary actor | Malicious user |
| *Threat Category* | *Injection attack* |
| *Threat pointcut* | 1. *POINTCUT register customer BF customer enters data JOINPOINT 1 && 2 && 4*<br>2. *POINTCUT make reservation BF customer enters data JOINPOINT 1 && 10*<br>*ADVICE AROUND POINTCUT ALL*<br>*EXECUTE BTF* |
| Precondition | 1. System has at least one registered user.<br>2. The system's network security is inadequate for detecting spoofed IP addresses<br>3. The webpage has web form.<br>4. The attacker spoofed the IP address. |
| Basic Threat Flow | 1. The MALICIOUS user PROVIDES SQLI VALUES IN input fields of the url.<br>2. The MALICIOUS user BYPASS the validation REQUEST TO server program.<br>3. The system executes the query provided in the url.<br>4. The system evaluates the query in the database.<br>5. The system VALIDATES THAT the query is successful.<br>6. The system SENDS the welcome message TO the MALICIOUS user. |
| Specific Alternative Threat Flow | RFS 5<br>1. DO<br>2. The system SEND the database error message DATA to the MALICIOUS user.<br>3. The MALICIOUS user EXPOLITES the database error message DATA from the system.<br>4. RESUME STEP 1<br>5. UNTIL the query is successful.<br>6. RESUME STEP 6 |
| Post condition | 1. The MALICIOUS user accesses the customer DATA without authorization |
| AFMUC ID | *AFMUC-04* |
| AFMUC name | *Mask a page* |
| Brief description | Obscure or hide the content of a web page from the user's view. |
| Primary actor | Malicious user |
| Threat category | *Sabotage attack* |
| Threat pointcut | *INTRODUCTION mask a page REPLACE register customer ‖ make reservation* |
| Precondition | 1. The system follows insecure coding practices.<br>2. The system has misconfigurations.<br>3. The attacker spoofed the IP address. |
| Basic Threat Flow | 1. The MALICIOUS user crafts a payload with code snippet.<br>2. The MALICIOUS user EXPOLITES the code injection vulnerability of server.<br>3. The MALICIOUS user GETS code FROM the code repository.<br>4. The MALICIOUS user PROVIDES REC VALUES IN the server code.<br>5. the MALICIOUS user SENDS a MALICIOUS payload TO the server program.<br>6. The MALICIOUS code BYPASS the secure code validation REQUEST TO server.<br>7. The system VALIDATES THAT the payload is valid.<br>8. The MALCIOUS user modify the server code with desired content. |
| Post condition | 1. The MALICIOUS user successfully defaces the target website. |
| AFMUC ID | *AFMUC-05* |
| AFMUC name | *Create malicious ads* |
| Brief description | creates malicious advertisements that appear legitimate. |
| Primary actor | Malicious user |
| Threat category | *Phishing attack* |
| Threaten pointcut | *INTRODUCTION mask a page ASSOCIATE TO register customer ‖ make reservation*<br>*AS INCLUDED USE CASE* |
| Precondition | 1. The system follows insecure coding practices.<br>2. The system has misconfigurations<br>3. The attacker spoofed the IP address. |

**Fig 9. Weaved AFMUC template to specify aspect threat.**

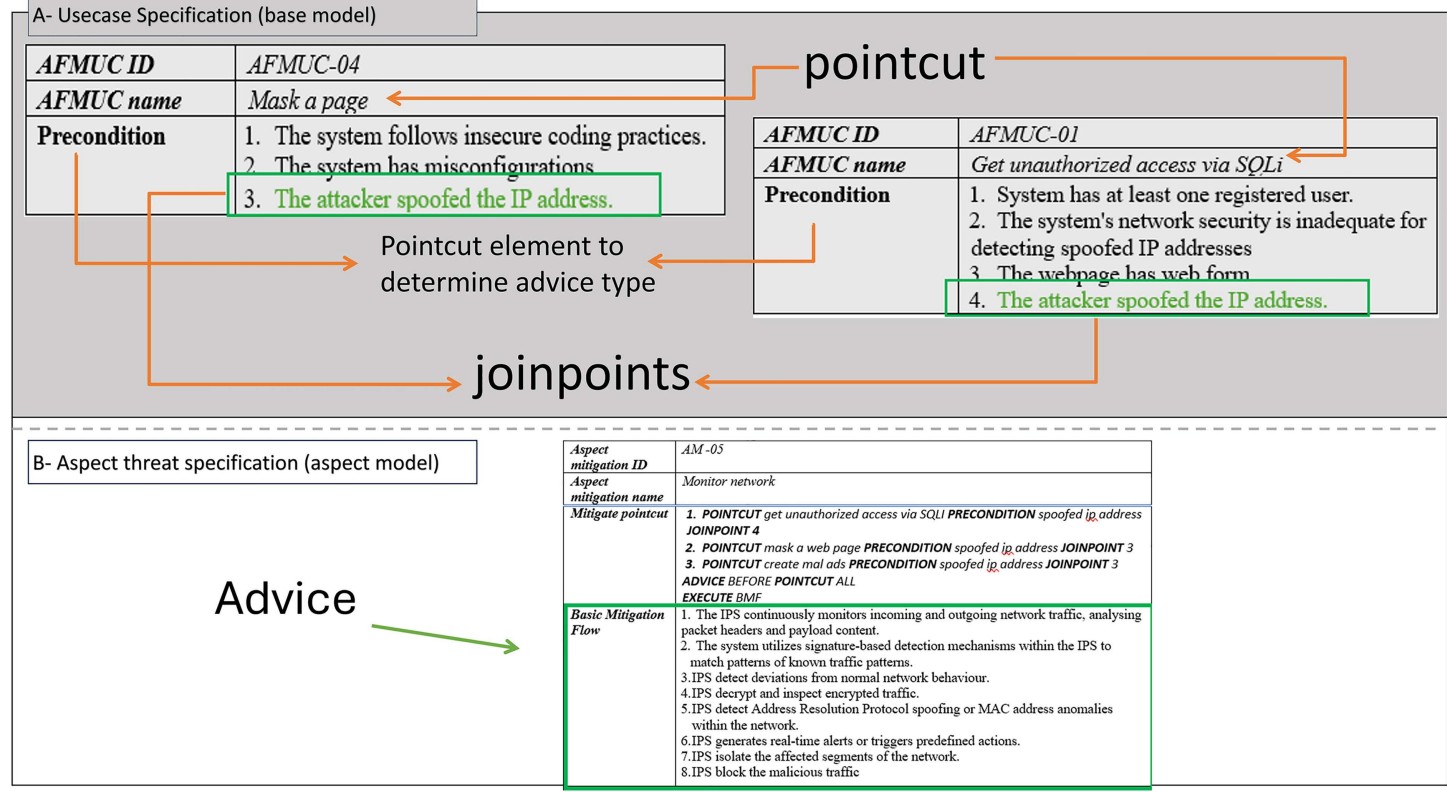

**Fig 10. Excerpt of aspect mitigation - pointcut, joinpoint, advice.**

application. In MFA, the sensitive information can be processed, stored, and shared by the customer while performing online transactions. The application's basic functional requirements include registering customer accounts, logging in, transferring money to various accounts (including bank, mobile, and CNIC), paying bills, loading mobile balances, purchasing mobile bundles, and generating account statements. Due to the involvement of financial transactions, the application is vulnerable to numerous threats. Attackers aim to exploit application vulnerabilities to access sensitive customer information by breaching databases, intercepting network communications, or launching phishing attacks. Additionally, attackers may seek to disrupt application services, rendering them inaccessible to customers. Their goal is to obtain confidential customer information or get financial gains from Company X. Given the time constraint, three significant use cases such as *register customer*, *login*, and *transfer money to bank account* are selected, which is threatened by five crosscutting threats (misuse cases) that pose significant threats such as unauthorized access, data modification, mask a web page to compromise services and phishing attack. One mitigation strategy, "*monitoring network*," is chosen to focus on, which crosscuts all the defined threats. The threats and mitigations are identified based on standards ISO27001 [36] and OWASP [35]. The excerpt of the MFA functional requirements, threats, and their mitigation is purposely selected to ensure the scope of study in a manageable time while completely covering the AFMUC approach. Completing the entire case study is out of the scope of this study

## Experimentation

A controlled experiment is conducted to evaluate the ambiguity of the proposed approach. The experiment is designed and reported according to the template provided by Wohlen et al. [38].

| Aspect mitigation ID | AM -05 |
|---|---|
| Aspect mitigation name | Monitor network |
| Brief Description | Involves analysing network traffic patterns, protocols, and data flows to identify abnormal behaviour, spoofed IPS and detect security threats |
| Mitigate pointcut | 1. **POINTCUT** get unauthorized access via SQLI **PRECONDITION** spoofed ip address **JOINPOINT** 4<br>2. **POINTCUT** mask a web page **PRECONDITION** spoofed ip address **JOINPOINT** 3<br>3. **POINTCUT** create mal ads **PRECONDITION** spoofed ip address **JOINPOINT** 3<br>**ADVICE** BEFORE **POINTCUT** ALL<br>**EXECUTE** BMF |
| Precondition | 1. The Intrusion Prevention (IPS) installed within the network infrastructure.<br>2. The network is operational and monitoring traffic. |
| Basic Mitigation Flow | 1. The IPS continuously monitors incoming and outgoing network traffic, analysing packet headers and payload content.<br>2. The system utilizes signature-based detection mechanisms within the IPS to match patterns of known traffic patterns.<br>3.IPS detect deviations from normal network behaviour.<br>4.IPS decrypt and inspect encrypted traffic.<br>5.IPS detect Address Resolution Protocol spoofing or MAC address anomalies within the network.<br>6.IPS generates real-time alerts or triggers predefined actions.<br>7.IPS isolate the affected segments of the network.<br>8.IPS block the malicious traffic |
| Alternate Mitigation Flow | RFS 2<br>1. The IPS mistakenly identifies legitimate traffic as spoofed.<br>2. The IPS drops legitimate packets.<br>RFS 5<br>1. The IPS does not identify the spoofed IP<br>2. The spoofed IP packets are not flagged and continue to the network.<br>3. The network is compromised |
| Post condition | 1. Spoofed IP addresses are detected and mitigated.<br>2. The network remains secure from the intrusion attempt. |

**Fig 11. Weaved AFMUC template to specify aspect mitigation.**

**Planning and design.** This section describes the planning and design of the controlled experiment.

• **Objective and motivation:** The experiment is conducted to perceive the viewpoint of the information security students about the unambiguity of the proposed approach for analyzing the crosscutting security threats and their mitigation.

• **Goal:** The goal is to assess how unambiguous the AFMUC is in modelling and specifying the aspect-oriented security threats and mitigations.

• **Research questions:** To address the goal of the experiment, the research question is formulated as:

RQ: Is the AFMUC unambiguous enough to model the crosscutting security threats and mitigations?

Therefore, this leads to the definition of the two-tailed hypothesis.

$H_0$: AFMUC is ambiguous in modeling crosscutting security threats and mitigations.

$H_1$: AFMUC is unambiguous in modeling crosscutting security threats and mitigations.

The ambiguity of AFMUC restriction rules to specify crosscutting security threats and mitigations, is measured in terms of understandability, applicability, restrictiveness, and error rate adopted from [31,39]. This experiment has four dependent

variables, namely *understandability*, *applicability*, *restrictiveness,* and *error rate*, and one independent variable as *AFMUC restriction rules*.

- **Dependent variables:**

i. The *understandability* is assessed based on a common understanding of the AFMUC constructs and restriction rules.

ii. The *applicability* is evaluated by determining whether the AFMUC constructs, and restriction rules are applicable to specifying the aspect threats and mitigation.

iii. R*estrictiveness* is evaluated to assess whether the AFMUC constructs, and restriction rules are challenging to specify aspect threats and mitigation.

iv. The *error rate* is used to measure incorrect implementation of the AFMUC restriction rule.

- **Independent variable:**

i. *AFMUC restriction rules* are unambiguous in terms of high understandability, applicability, low restrictiveness and error rate.

- **Population and sampling strategy:** The target population of the experiment consists of university students as participants from an information security program. The sample of 24 students is selected using a non-probabilistic convenience sampling technique [40]. Considering this, the students are chosen based on ease of access, having studied courses such as information security, cybersecurity, and secure software design and development. The students are familiar with the threats, mitigations, UML designs, and misuse cases. However, students have no prior understanding of aspect-oriented modeling. To address potential bias, a training session is conducted to familiarize the subjects with AOM before the experiment. The rationale for selecting these students is to find samples that could quickly be trained to use the AOM approach over a short time. The students are recruited for two days and motivated to attend the experiment by being graded for participation. On March 19, 2024, a comprehensive training session is held and completed. On the second day, March 26, 2024, a controlled experiment is conducted to apply and evaluate the proposed approach. Both sessions lasted three hours. Before engaging in the experiment, students are asked to sign a consent form indicating their willingness to utilize their experiment results for research purposes. Additionally, the experiment plan had undergone a thorough review and received approval from the Research Ethics Committee of the Computing Department of Riphah University before the execution of the experiment. Subjects actively involved in the experiment expressed their agreement by signing the consent form, confirming their understanding and approval to utilize the obtained data for research purposes. This process ensured a transparent and ethically sound approach to obtaining participants' informed consent.

A questionnaire-based evaluation is selected for data collection on the reasoning by Ardalan et al. [19] that constitutes three things: (1) the sample population is not too large, (2) the responses are required on the day of the experiment, and (3) quantitative data is required.

- **Data collection:** The ambiguity is determined by using four metrics such as understandability, applicability, restrictiveness, and error rate. Quantitative data is collected for each of the four metrics. Understandability is measured by a dichotomous scale (Yes/No) while applicability and restrictiveness are assessed using a four-point Likert scale. Also, mathematical formulas are designed to calculate the error rate on a scale of 0 and 1.

- **Questionnaire design:** The quantitative questionnaire is designed to address the research question by focusing on the dependent variables. Each questionnaire is designed to obtain responses for AFMUC constructs and restriction rules. The questionnaire guidelines are adopted from Yue et al. [31,39]. Table 4 represents the questionnaire consisting of three closed-ended questions on the *understandability*, *applicability,* and *restrictiveness* of AFMUC. Question 1 is used to measure the *understandability* of the approach on a scale of yes/no. Each rule is evaluated to determine whether the subject understands the restriction rule and shares a common understanding. Question 2 contains the question regarding *applicability* of the approach. It defines whether the subjects can

**Table 4. Questionnaire to evaluate ambiguity of AFMUC in terms of understandability, applicability and restrictiveness.**

| Question/ Rule # | AFMUC Restriction Rule | Understand-ability | | Applicability | | | | Restrictiveness | | | |
|---|---|---|---|---|---|---|---|---|---|---|---|
| | | Question 1: I understood the restriction rule and was able to apply it properly | | Question 2: The restriction rule was straight-forward to apply | | | | Question 3: Did you feel that the restriction rule was too restrictive to apply | | | |
| | | Yes | No | Completely agree | Generally, agree | Generally, disagree | Completely disagree | Completely agree | Generally, agree | Generally, disagree | Completely disagree |
| R 1. | BF | | | | | | | | | | |
| R 2. | PRECONDITION | | | | | | | | | | |
| R 3. | POSTCONDITION | | | | | | | | | | |
| R 4. | BTF | | | | | | | | | | |
| R 5. | BMF | | | | | | | | | | |
| R 6. | POINTCUT<name> SUBSET<selection constraint> | | | | | | | | | | |
| R 7. | JOINPOINT <sequence no> | | | | | | | | | | |
| R 8. | ADVICE<type> POINTCUT <sequence no> EXECUTE <flow> | | | | | | | | | | |
| R 9. | BEFORE | | | | | | | | | | |
| R 10. | AFTER | | | | | | | | | | |
| R 11. | AROUND | | | | | | | | | | |
| R 12. | && | | | | | | | | | | |
| R 13. | \|\| | | | | | | | | | | |
| R 14. | ALL | | | | | | | | | | |
| R 15. | INTRODUCTION <name> ASSOCIATE TO <use case> AS <dependency> | | | | | | | | | | |
| R 16. | INTRODUCTION <name> REPLACE <use case> | | | | | | | | | | |

properly apply the AFMUC restriction rules for specifying the aspect threats and their aspect mitigations. It is evaluated on a four-point Likert scale. Lastly, question 3 is used to evaluate the *restrictiveness* of the proposed approach, which assesses whether the subject finds the restriction rule challenging to specify. It is also evaluated on a four-point Likert scale.

This questionnaire aimed to gather their subjective opinions on the proposed AFMUC approach, evaluating their perceptions of understandability, applicability, and restrictiveness. The quiz assessment is designed to ensure their understanding of the proposed approach to avoid any potential subjectivity bias in obtaining the data for each measure. The complete quiz is given in "S1 Appendix". The quiz questions are designed for each aspect. There are three sections for each question, Section A consists of questions regarding the pointcut expression, including joinpoints. Section B defines the question regarding the advice, and Section C presents a general question. Regarding the aspect element introduction, Section A addresses questions on introduction expression, and Section B consists of a general question. All questions

are closed-ended questions, structured as multiple-choice questions, except one open-ended general question where students are asked to justify their choice, whether they experienced any uncertainty in identifying threats or mitigations. All multiple-choice questions contain one correct answer except those marked with three asterisk symbols (***). The justification provided by subjects for each answer is carefully analyzed to assess their understanding of the respective aspects. Total, there are ten multiple-choice questions covering aspects threats and mitigation, and six general questions. The responses to the quiz questionnaire are used to validate the data collected, but they are not analyzed to test the experimental hypothesis.

• **Error rate measure:** For the measure of error rate, the raw data is collected from the AFMUC aspect threats and mitigation template attempted by each student. The error rate measures the inaccurate application of the restriction rule. It is measured on a scale of 0 and 1. Table 5 describes the units to collect the raw data in the form of mistakes made by students in each aspect. These units measure the individual error count for each AFMUC restriction rule presented in Table 6. And finally, the sum of error rate for each restriction rule, aggregated across all students, is calculated using the formula Error Rate$_r$, as defined in equation 1. All these units and formulas are adopted from Yue et al. [31,39] and modified according to AFMUC restriction rules. For example, the error rate for R1 is calculated as the average of ($N_{missed}/N_{UC}$). It captures the rate of occurrences where the BF keyword should be applied, but it is missing. The variable ($N_m$) calculates total mistakes made when applying the keyword **BF**. The formula ($N_m/N_{BF}$) calculates the frequency of keyword BF being misused, applied in an inappropriate context, or used when it should not be. The specifications of the error rate for each rule are specified in Table 6, along with the description of variable ($N_m$) for measuring the errors for each rule. "NA" represents the measure is not applicable for restriction rule in a specific aspect.

The formula for calculating the error rate of each restriction rule is further defined as follows:

**Table 5. Metrics to calculate raw data for each aspect.**

| Measure | Specification in a single aspect |
| --- | --- |
| $N_{UC}$ | Total number of use cases |
| $N_{missed}$ | Number of instances where the keyword should be applied, but is not |
| $N_{aspect}$ | Total number of aspects |
| $N_V$ | Number of instances where the rule is violated. |
| $N_{BP}$ | Total number of basic flows |
| $N_{PRECONDITION}$ | Total number of preconditions |
| $N_{POSTCONDITION}$ | Total number of postconditions |
| $N_{BTF}$ | Total number of basic threat flows |
| $N_{BMF}$ | Total number of basic mitigation flows |
| $N_{POINTCUT}$ | Total number of pointcuts |
| $N_{JOINPOINT}$ | Total number of joinpoints |
| $N_{ADVICE}$ | Total number of advice |
| $N_{BEFORE}$ | Total number of before |
| $N_{AFTER}$ | Total number of after |
| $N_{AROUND}$ | Total number of around |
| $N_{\&\&}$ | Total number of && |
| $N_{\|\|}$ | Total number of \|\| |
| $N_{ALL}$ | Total number of all |
| $N_{D\_INTROD}$ | Total number of dependency introductions |
| $N_{INTRODUCTION}$ | Total number of introductions |

**Table 6. Measure of the error rate for each restriction rule as aspect threat and aspect mitigation.**

| Rule # | Restriction Rule | Measure (aspect misuse case) | Measure (aspect mitigation) | $N_m$ |
|---|---|---|---|---|
| R 1. | BF | $((N_{missed}/N_{UC})+(N_m/N_{BF}))/2$ | NA | number of occurrences where the keyword is applied but in a wrong situation or should not be applied |
| R 2. | PRECONDITION | $((N_{missed}/N_{UC})+(N_m/N_{PRECONDITION}))/2$ | $((N_{missed}/N_{MUC})+(N_m/N_{PRECONDITION}))/2$ | number of occurrences where the keyword is applied but in a wrong situation or should not be applied |
| R 3. | POSTCONDITION | $((N_{missed}/N_{UC})+(N_m/N_{POSTCONDITION}))/2$ | $((N_{missed}/N_{MUC})+(N_m/N_{POSTCONDITION}))/2$ | number of occurrences where the keyword is applied but in a wrong situation or should not be applied |
| R 4. | BTF | $((N_{missed}/N_{aspect})+(N_m/N_{BTF}))/2$ | NA | number of occurrences where the keyword is applied but in a wrong situation or should not be applied |
| R 5. | BMF | NA | $((N_{missed}/N_{aspect})+(N_m/N_{BMF}))/2$ | number of occurrences where the keyword is applied but in a wrong situation or should not be applied |
| R 6. | POINTCUT<name> SUBSET <selection constraint> | $((N_{missed}/N_{UC})+(N_m/N_{POINTCUT}))/2$ | $((N_{missed}/N_{MUC})+(N_m/N_{POINTCUT}))/2$ | number of occurrences where the keyword is applied but in a wrong situation, or incorrect (mis)use case name or selection constraint to select the (mis)use case subset or should not be applied |
| R 7. | JOINPOINT <number> | $((N_{missed}/N_{UC})+(N_m/N_{JOINPOINT}))/2$ | $((N_{missed}/N_{MUC})+(N_m/N_{JOINPOINT}))/2$ | number of occurrences where the keyword is applied but in a wrong situation, or incorrect selection of join points in (mis)use case elements, or should not be applied |
| R 8. | ADVICE<type> POINTCUT <sequence no> EXECUTE <flow> | $((N_{missed}/N_{aspect})+(N_m/N_{ADVICE}))/2$ | | number of occurrences where the keyword is applied but in a wrong situation, or with incorrect flow, or pointcuts or should not be applied |
| R 9. | BEFORE | $((N_{missed}/N_{aspect})+(N_m/N_{BEFORE}))/2$ | | number of occurrences where the keyword is applied but in a wrong situation or should not be applied |
| R 10. | AFTER | $((N_{missed}/N_{aspect})+(N_m/N_{AFTER}))/2$ | | number of occurrences where the keyword is applied but in a wrong situation or should not be applied |
| R 11. | AROUND | $((N_{missed}/N_{aspect})+(N_m/N_{AROUND}))/2$ | | number of occurrences where the keyword is applied but in a wrong situation or should not be applied |
| R 12. | && | $N_v/N_{\&\&}$ | | NA |
| R 13. | \|\| | $N_v/N\|\|$ | | NA |
| R 14. | ALL | $N_v/N_{ALL}$ | | NA |
| R 15. | INTRODUCTION <name> ASSOCIATE TO <use case> AS <dependency> | $((N_{missed}/N_{aspect})+(N_m/N_{D\_INTRO}))/2$ | NA | number of occurrences where a keyword is applied but in a wrong situation, or with incorrect aspect, or use case name, or dependency, or should not be applied |
| R 16. | INTRODUCTION <name> REPLACE <use case> | $((N_{missed}/N_{aspect})+(N_m/N_{INTRO}))/2$ | NA | number of occurrences where the keyword is applied but in a wrong situation, or with incorrect aspect or use case name or should not be applied |

$$ErrorRate_r = \frac{\sum_{S=1}^{|S|}\left(\sum_{aspect=1}^{|aspect|} ErrorRate_{aspect}\right)}{aspect \times S}$$

(1)

Error Rate$_r$ represented the sum of errors made by all the students S for a particular AFMUC restriction rule across all defined aspects ErrorRate$_{aspect}$. In other words, each student evaluated the application of a given AFMUC restriction rule across all defined aspects, and any mistake made during implementation is measured as an error. The sum of these errors across all students and all aspects is used to calculate the overall error rate for that rule. This metric provides a quantitative measure of incorrect implementation of the AFMUC restriction rule.

• **Statistical test:** A Spearman correlation statistical test [41] is used to measure the strength and relationship between understandability, applicability, restrictiveness, and error rate. The proposed approach is assumed to be highly understandable and applicable with less restrictiveness while reflecting a low error rate.

## Experiment preparation and execution

The following are the steps outlined to prepare and execute the control experiment.

**Experiment objects.** The experiment objects include the MFA functional requirements specifications as filled use cases using the RUCM specification template, and threat and mitigation specifications using partially filled AFMUC misuse case and mitigation templates. The evaluation questionnaire and quiz are also provided to participants.

These participants are provided with six partially specified aspect documents in AFMUC, five aspect misuse cases, and one aspect mitigation case. Three MFA use cases are modelled using RUCM, aiming to ensure complete coverage of the AFMUC constructs and restriction rules.

**Process of experiment.** The participants are required to document aspect-oriented misuse cases using AFMUC. To facilitate this, they are given partially filled AFMUC misuse cases and mitigation cases. These cases included descriptions for elements, such as *AFMUC ID*, *AFMUC name*, *brief description*, *primary actor*, *threat category*, *precondition*, *basic threat flow*, *alternate threat flow*, and *post-condition*. However, the students are required to specify the *threat pointcut* section. Similarly, the AFMUC mitigation template is also provided with descriptions filled for the *Aspect Mitigation ID*, *Aspect mitigation name*, *brief description*, *basic mitigation flow*, *alternate mitigation flow*, *precondition,* and *postcondition,* except the *mitigate pointcut* section. This approach is adopted to focus students' attention on the core concepts of AFMUC by just specifying aspect threats and aspect mitigation.

**Piloting of the experiment.** To identify and address potential issues and ambiguities in the experiment procedure, the experiment is piloted on two undergraduate-level students and two faculty members with their background knowledge in cybersecurity, information security, secure software design, and development. The responses are carefully evaluated. Based on their feedback and discussion with students and faculty members, the experiment procedure is refined to ensure the quality of the experiment.

**Training of participants.** The success of the actual experiment is dependent on the effectiveness of the training session, making it crucial for students to receive comprehensive training on the new approach. A training session is conducted one week before the experiment to idealize the students' background knowledge and skills. The purpose of this training session is to familiarize the subjects with the basic concepts of AFMUC approach and its artifacts. Subjects engaged in a continuous and interactive three-hour training session, focused on activity-based learning to provide a practical understanding of concepts. The session includes the revision of threat and mitigation modeling using misuse cases and training of essential AOM constructs (i.e., pointcut, joinpoint, advice, introduction). In this session, the explanation of approach is given to the students on the EU-Rent Car [37] case study as an illustrative example. To further enhance students' preparedness, a home-based graded assignment is designed to help them practice applying the AFMUC and its restriction rules, enabling them to specify aspect-oriented security threats and their corresponding mitigations effectively. Ensuring that they had adequate time to review and understand the basic concepts. Detailed feedback is provided on an assignment before the experiment to address misunderstandings and strengthen their background knowledge. This is planned intentionally because the subjects do not know the AOM used in proposed AFMUC approach. This is carefully maintained to allow the students adequate time to assimilate the training content, including the AOM and restriction rules. Since this is their first experience with it, they need a basic understanding before participating in the experiment.

**Execution of experiment.** In this session, students are asked to perform a task on an MFA case study in a controlled laboratory during a three-hour session. All 24 students attended the session, and they are required to remain in the session until the task is completed. Students are instructed to specify the five aspect threats using the AFMUC misuse case template and one aspect mitigation using the AFMUC mitigation template. And lastly students attempted the evaluation questionnaire and a quiz.

## Results and analysis

This section describes the quantitative analysis of results collected from the controlled experiment. Four out of twenty-four responses are excluded from the results analysis due to being empty or incomplete. One participant is a foreign student

with a nominal understanding of English, facing a language barrier in understanding and applying the restriction rule. Therefore, this response is excluded from the analysis. The remaining three responses are excluded due to being incomplete. All the other participants completed the task within the three-hour session. In total, twenty responses are utilized for result analysis.

### Ambiguity in aspect threat and mitigation specification

The objective of the research question is to evaluate the ambiguity of the proposed approach in terms of understandability, applicability, restrictiveness, and error rate. For this purpose, each restriction rule is evaluated using four quantitative measures: understandability, applicability, restrictiveness, and error rate. The first three measures are used to determine the subjective opinion of the students, while the error rate gives the objective measure to collect the errors while applying the rules by students. The results are compiled and presented individually for each measure. Also, a statistical Spearman's rank correlation coefficient test is applied to assess the correlation between these four measures. The data collected for each metric is provided in "S2 Data".

**Understandability.** Understandability is measured based on a subjective opinion of students for each restriction rule. As shown in Fig 12, twelve out of sixteen rules are considered highly understandable with a threshold frequency rate of 90% or above. The rules R3, R4, R6, and R13 received a frequency of 90%. Rules R1, R5, R9, R10, and R14 are rated at 95%. Meanwhile, rules R2, R11, and R12 are rated completely understandable with a 100% frequency rate. In contrast, rules R7 (JOINPOINTS) and R8 (ADVICE) received an 85% score, which is slightly understandable. However, the R15 (70%) and R16 (80%) got the lowest score, with R15 representing the least understandable rule. It is observed that R15 and R16 are related to the "introduction" construct in AFMUC, a concept that students find difficult to grasp. This highlights a need to put more training effort into these rules.

In general, it is observed that most of the restriction rules are understandable up to a sufficient level by the students.

**Applicability.** Each rule's applicability rate is determined according to how students believe it should be applied using a four-point Likert scale from completely agree as "1" to completely disagree as "4". In Fig 13 shows the

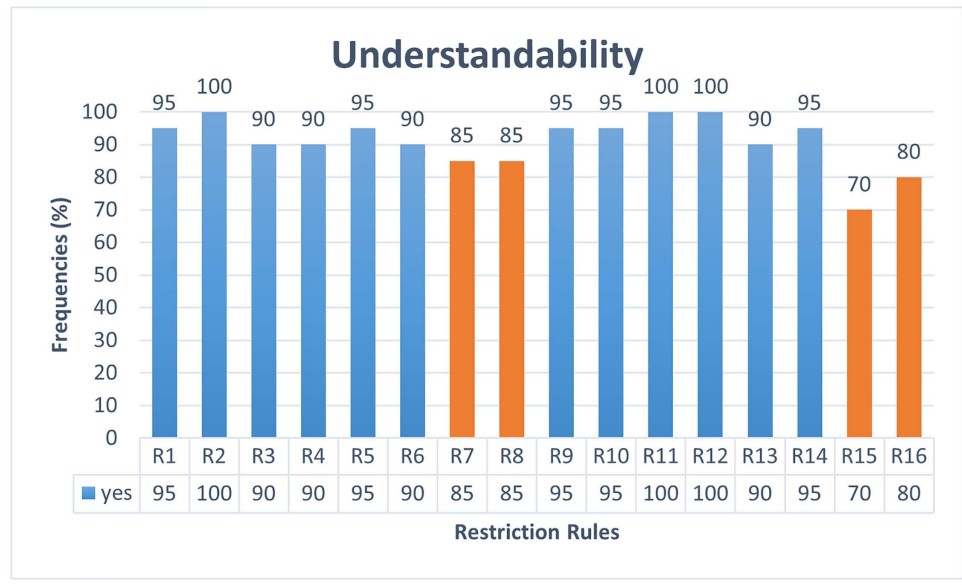

**Fig 12. Understandability of AFMUC restriction rules.**

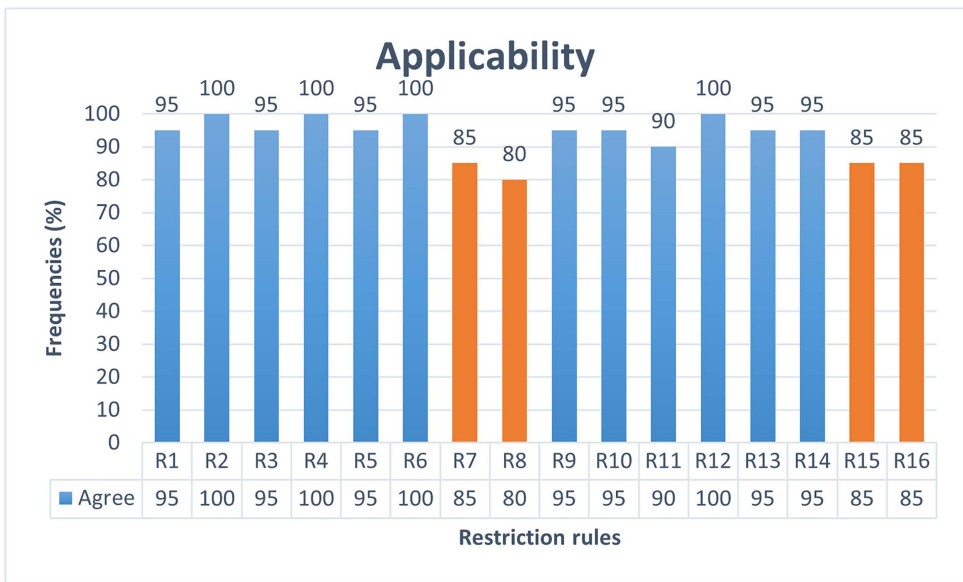

**Fig 13. Applicability of AFMUC restriction rules.**

analysis of high applicability based on the frequency of responses rated as "agree" for the restriction rules when the threshold score is 90% or above. And disagree when the score is below 90%. The results show that twelve out of sixteen rules are highly applicable, with most students showing high applicability. Within these, rule R11 gets 90%; rules R1, R3, R5, R9, R10, R13 and R14 get a 95% rating. while R2, R4, R6, and R12 are considered fully applicable because they all have 100% agreement. However, rules R7, R8, R15, and R16 receive an applicability rating below the 90% threshold. Each rule, R7, R15, and R16, gets an applicability score of 85%. With 80%, R9 is considered the least applicable. Although the applicability of these rules is slightly lower than others, the difference is not significant enough to suggest they are inapplicable. It is observed that rules R7, R8, R15, and R16 received a lower rate not only in applicability but also in understandability. This indicates that they are more challenging to understand and thus relatively difficult to apply. Intensive training or automated tool support could improve the understanding and applicability of these rules.

**Restrictiveness.** Fig 14 shows the restrictiveness of the rules, which shows the subjective measure of the students regarding the difficulty and challenges in specifying the aspect-oriented threats and mitigation. The measure is assessed on the scale of four-point Likert scale, "1" for completely agree and "4" for completely disagree. The results are analyzed on the frequency of responses for "agree" which tells the perceived restrictiveness. The threshold for low restrictiveness is defined as 20% or less showing "agree" and "disagree" when the frequency is greater than 20%. The results show that twelve out of sixteen rules are highly restrictive, with scores above 20%. This includes rules R1, R3, R5-R11, R13, and R15-R16. Among these, rule R8 has the highest restrictiveness rate at 45%, indicating that applying the *Advice* is challenging for students. Rules R15 and R16 exhibit high restrictiveness with a rate of 40%. Notably, these three rules, R8, R15, and R16, have also received a lower rate of understandability and applicability. The rules R6 (35%), R13 (30%), R1, R3, R5, R7, R9, R10, and R11 each show the restrictiveness rate of 25%, which may be considered less challenging given their high rating for understandability and applicability. In contrast, four rules, R2, R4, R12, and R14 are identified as less restrictive. These rules are also most understandable and applicable, indicating that students find them easy to understand and apply.

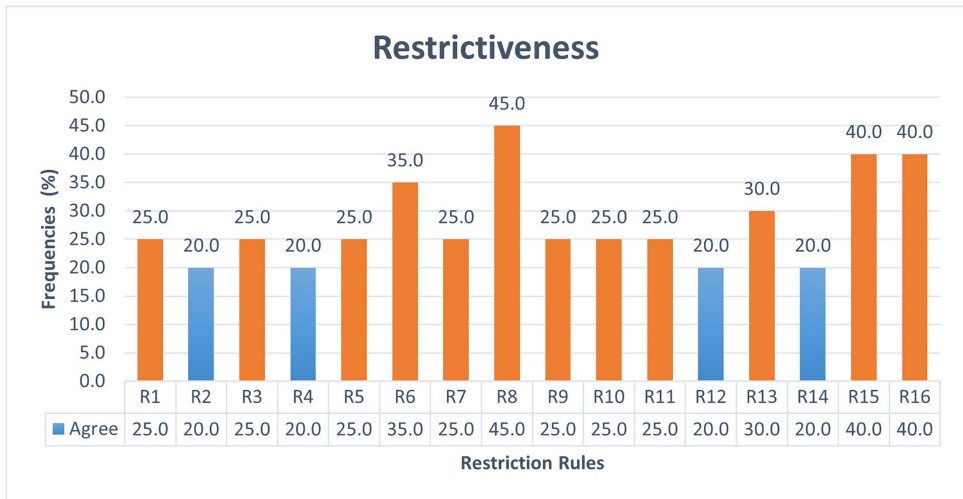

**Fig 14. Restrictiveness of AFMUC Restriction Rules.**

Overall, 80% of the students find the rules too restrictive for specifying aspect-oriented threats and their mitigation. This is likely because they applied the rules for the first time and have only recently been introduced to aspect-oriented modelling techniques, which are new to them. This unfamiliarity has made it challenging for them to specify aspect-oriented threats and their mitigation effectively.

**Error rate.** Error rate represents the incorrect implementation of the restriction rules while specifying the aspect-oriented threats and their mitigation. The rule is considered to have a high error rate when its score is ≥ 5%. In Fig 15, the results show that ten out of sixteen restriction rules attain a high error rate. The highest error rates are R11 and R6, at 11.6% and 10%, respectively. The high error rate for R11 is due to the misuse of the AROUND advice type rule, often applied in situations where it is not required. Similarly, R6 has a high error rate because the pointcut expression contains

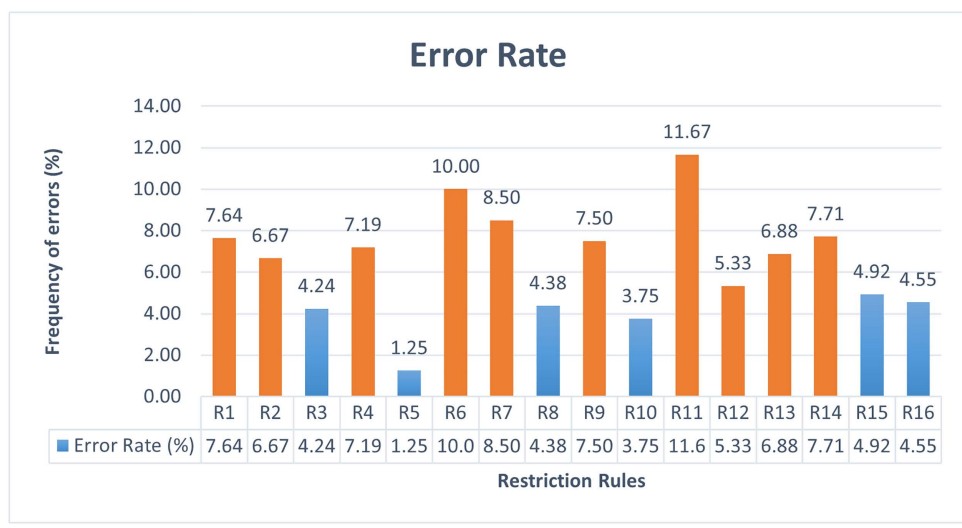

**Fig 15. Error Rate of AFMUC Restriction Rules.**

three different logical expressions inside: (1) specifying the name of the use case or misuse case where the aspect threat or mitigation aspect pointcuts, (2) selecting elements such as BF or BTF in the relevant case, and (3) selecting the constraint to identify the attack surface in the use case or the mitigation surface in the misuse case. It is observed that the high error rate for R6 is due to not completing the expression or specifying the expression in the wrong situation. It is also analyzed that ten rules with high error rates, R1, R2, R4, R6, R7, and R9, R11–R14, involve key constructs of aspect-oriented modeling, such as specifying pointcut R6 (10%) and joinpoint R7 (8.50%). It shows a higher error rate in identifying joinpoints in the basic flow R1 (7.64%), and precondition R2 (6.67%), and applying the type of AROUND advice R11 (11.6%), and BEFORE advice R9 (7.50%), respectively. The advice threat behavior (BTF), as R4 also shows, has a higher error rate of 7.19%. Furthermore, using logical operators R12, R13, or ALL R14 to select joinpoints also contributed to higher error rates at 5.33%, 6.88%, and 7.71%, respectively. These quantitative findings suggest that students face challenges applying these rules specific to AOM constructs, such as pointcut, joinpoint, and advice. Additionally, it is also analyzed that rules R2, R4, R12, and R14 are perceived as highly understandable, applicable, and less restrictive to use however they present challenges during implementation in actual scenario. It is believed that the high error rates for these essential AOM constructs stem from limited exposure and experience with aspect-oriented modeling. Providing tool support or further training on these rules helps lower the error rate.

In general, it is observed from the results that very few restriction rules are correctly implemented, such as R3, R5, R8, R10, R15, and R16, with low error rate. These rules include introduction expression (R15, R16), applying AFTER (R10) advice at the POSTCONDITION (R3) with correct advice expression (R8), and implementing mitigation flow (R5) to threats.

## Statistical analysis

This section establishes the significance of results through statistical analysis. It analyzes the correlation between the understandability, applicability, restrictiveness, and error rate. It is expected that the rules that are easy to understand and apply should be less restrictive and have a lower error rate. For this a statistical test Spearman non-parametric correlation test [41] is performed between each pair of understandability, applicability, restrictiveness, and error rate. Statistical significance is measured using a p-value threshold of ≤ 0.05. Table 7 shows the statistical relationship between the three measures as understandability, applicability, and restrictiveness. It is observed that there exists a positive monotonic relationship between these three variables. Understandability has a meaningful and statistically significant positive relationship with both applicability and restrictiveness with the p-values as 0.022 and <0.001, respectively. This indicates that the AFMUC restriction rules perceived as understandable are also considered as more applicable and less restrictive to use. Similarly, a significant relationship between applicability and low restrictiveness (p = 0.003) also reinforce that AFMUC rules perceived as easier to apply are also less restrictive. These significant relationships between understandability, applicability and restrictiveness, each with p-values below the threshold (p value = 0.05) rejects the null hypothesis and confirms that AFMUC restriction rules are unambiguous to specify the crosscutting security threats and mitigations.

**Table 7. Summary of the spearman correlation test.**

| Measure | Measure By | Spearman p | Prob > |p| |
|---|---|---|---|
| High Applicability | Understandability | 0.566 | 0.022 |
| Low Restrictiveness | Understandability | 0.779 | <0.001 |
| Error Rate | Understandability | 0.174 | 0.52 |
| High Applicability | Low Restrictiveness | 0.696 | 0.003 |
| Error Rate | High Applicability | 0.13 | 0.63 |
| Error Rate | Low Restrictiveness | 0.096 | 0.725 |

On the other hand, the correlation between error rate and other measures, e.g., understandability (p-value = 0.52), applicability (p value = 0.63) and low restrictiveness (p = 0.725) is weak and statistically insignificant. This indicates that despite being perceived as unambiguous AFMUC rules may still lead to implementation errors. The reason behind this could be that the results are not consistent with all the students, some students have got the maximum understanding of the rules and are able to apply them while making no mistakes or very few. However, some of the students do not have enough experience to apply the rules correctly; those students may have found the rules understandable and applicable with less restrictiveness but when they specify the rules, they make mistakes.

Table 8 shows the integrated result of all four measures to manually analyze the relationship between understandability, applicability, restrictiveness, and error rate. It is worth noting that rules R2, R4, R12 and R14 show high understanding and applicability (90%−100%) and are least restrictive (20%), but still have non-negligible error rate (6.67%, 7.19%. 5.33%, 7.71% respectively) when implemented. The rules R3, R5 and R10 generally scored above 90% in understandability and applicability and low error rate as 4.42%, 1.25% and 3.75% respectively. However, they are slightly restrictive, each with a restrictive rate of 25%. Compared to other rules, R2, R4, R12, R14, R3, R5 R10 can be considered among the most unambiguous AFMUC rules. These rules exhibit consistently high understandability and applicability along with slight restrictiveness (R3, R5, and R10) or slightly elevated error rate (R2, R4, R12 and R14). Which can be further enhanced to reduce implementation errors or make it less restrictive. The rules R1, R6, R9, R11, and R13 also exhibit high understandability and applicability (90% and above), but they also show high restrictiveness and a higher error rate. Therefore, these rules can be categorized as moderately unambiguous AFMUC restriction rules, as they are generally understandable and applicable but are challenging and tend to result in errors during implementation. The rules R8, R15, and R16 can be categorized as highly ambiguous, as they contrast a low error rate (less than 5%) with poor results on the other three measures such as understandability, applicability below 90% and restrictiveness above 20%. This indicates that students have perceived these rules as difficult to understand, less applicable, and challenging, but they applied these rules correctly in accordance with no or few mistakes. And lastly, R7 stands out as the highest ambiguous rule, exhibiting poor results across all four evaluation measures of ambiguity. It shows low understandability (85%), low applicability (85%), is considered restrictive (25%), and has a high error rate (8.5%). This indicates that students face consistent difficulty in understanding and implementing the R7.

**Table 8. Comparative analysis of understandability, applicability, restrictiveness and error rate.**

| Rules | Understandability | High Applicability | High Restrictiveness | Error Rate |
|---|---|---|---|---|
| R1 | 95 | 95 | 25 | 7.64 |
| R2 | 100 | 100 | 20 | 6.67 |
| R3 | 90 | 95 | 25 | 4.24 |
| R4 | 90 | 100 | 20 | 7.19 |
| R5 | 95 | 95 | 25 | 1.25 |
| R6 | 90 | 100 | 35 | 10.00 |
| R7 | 85 | 85 | 25 | 8.50 |
| R8 | 85 | 80 | 45 | 4.38 |
| R9 | 95 | 95 | 25 | 7.50 |
| R10 | 95 | 95 | 25 | 3.75 |
| R11 | 100 | 90 | 25 | 11.67 |
| R12 | 100 | 100 | 20 | 5.33 |
| R13 | 90 | 95 | 30 | 6.88 |
| R14 | 95 | 95 | 20 | 7.71 |
| R15 | 70 | 85 | 40 | 4.92 |
| R16 | 80 | 85 | 40 | 4.55 |

However, these unusual results may stem from the fact that aspect-oriented modeling and restriction rules are relatively new concepts to the students, leading to challenges in both understanding and implementation.

## Discussion

The objective of this study is to identify crosscutting security threats and mitigations early in the software development process. The misuse case specification template is extended into the AFMUC, which represents aspect-oriented essential constructs such as pointcut, joinpoint, advice, and introduction to specifying these crosscutting security threats and mitigations. A discussion of the study results considering the research questions as follows:

RQ: Is the AFMUC unambiguous enough to model the crosscutting security threats and mitigations?

The findings indicate that the proposed approach, AFMUC, unambiguously specifies crosscutting threats and their mitigations. The approach is easy to understand, applicable, and less restrictive in specifying aspect threats and mitigations. However, the observations of the error rates reveal that few essential aspect-oriented constructs specifically pointcut, joinpoint, and the application of before, after, and around advice, exhibited high error rates. This suggests that while the AFMUC is initially understandable and adaptable, its practical implementation in real scenarios presents challenges, making it difficult to accurately specify these essential concepts within the AFMUC model.

Reflecting the purpose of proposed approach, the AFMUC approach is unique and novel, as the existing literature does not provide evidence of utilizing misuse case and mitigation specification models for aspect threats and their mitigation with pointcut, joinpoint, advice, introduction, and weaving. Additionally, AFMUC is designed to be generic as it provides a reusable and extensible framework that can be adaptable to diverse application domain. Furthermore, the approach is scalable, enables the effective handling of large system with growing functionality and number of crosscutting threats and their corresponding mitigations. AFMUC modular design supports scalability by allowing to model crosscutting threats and mitigation independently as aspect threats and aspect mitigations and then woven to use cases and misuse case respectively. This scalability is particularly advantageous for security analysts, as it allows for an unambiguous and manageable specification of crosscutting security requirements within larger systems, enhancing overall application security. Furthermore, it benefits the security testers to precisely identify all attack surfaces and mitigate them by identifying corresponding mitigation surfaces.

### Comparison of existing (mis)use case approach

This section compares the proposed approach with existing work that specifying the specify the crosscutting concerns using use case Since proposed work AFMUC focuses on the modelling of crosscutting security concerns through misuse case model. Tables 9 and 10 compared with two dimensions: (1) features of aspect oriented supported by use cases such as pointcut, joinpoint, advice and its type, introduction and weaving. (2) features of (mis)use cases supporting by the aspect oriented approach such as precondition, basic flow, alternate flow, postcondition and dependency. Based on the above comparison criteria, eleven existing studies [10,14–21] address the aspect oriented features in the context of use cases. Table 9 characterizes these works with respect to their coverage of important aspect oriented modelling constructs including aspect, pointcut, joinpoint, advice, types, introduction and weaving also characterizes the model and requirements that are specified as crosscutting concerns. For instance, Table 9, the approach presented in [18] and [19]only supports modelling crosscutting security concerns using aspect oriented construct as pointcut, joinpoint and advice (indicated by +sign) in use case specification model. These constructs are also found in other approaches [10,14,17,20,21] but they are used to specify the crosscutting functional requirements [20,21]or used use case diagram instead of specification [10,14,17]. The modelling constructs that are not supported by these approaches are indicated by a – sign. Certain approaches [15,16] address the modelling of crosscutting NFR(e.g., security) using stereotypes but are not supported by AOM constructs. In related work, the existing misuse case templates [7,8] also compared with proposed AFMUC. Table 9

Table 9. Comparative analysis of AOM features supported by (mis)use cases.

| Reference | Model | Requirements | Aspect | Pointcut | Joinpoint | Advice | Before | After | Around | Introduction | Weaving |
|---|---|---|---|---|---|---|---|---|---|---|---|
| [10,14] | Use case diagram | Functional | Use case | + | + | + | – | – | – | – | – |
| [15,16] | Use case diagram and specification | NFR | Use case stereotypes | – | – | – | – | – | – | – | – |
| [17] | Use case diagram | NFR (performance, logging, authentication | Use case as aspect block | + | + | + | + | + | + | – | – |
| [18] | Use case specification | Security | Use case | + | + | + | – | – | – | – | – |
| [19] | Use case specification | Security | Aspectual use case | + | + | + | – | – | – | – | – |
| [20] | Use case specification | Functional | Aspectual use case | + | + | + | – | – | – | – | + |
| [21] | Use case specification | Functional | Use case | + | + | + | + | + | + | – | – |
| [7,8] | Misuse case specification | Security | Not supported | – | – | – | – | – | – | – | – |

suggests that no misuse case approach is exclusively specify the crosscutting security concerns (e.g., threats and mitigation). Table 10 provides information on the use case features (e.g., precondition, basic flow, alternate flow, postcondition, dependency) use by each approach for modeling aspect oriented constructs, whether the pointcut selects the joinpoints in any of these Use case elements. Table 10 suggests that in [18] and [19] the AOM constructs are used to specify crosscutting security concerns at use case precondition, basic flow, alternate flow and post condition. And related work [21] applied the maximum AOM constructs on all use case elements such as precondition, basic flow, alternate flow and post condition but for specification of crosscutting functional requirements.

Based on Tables 9 and 10, it is concluded that the modelling of crosscutting security concerns using AOM constructs is significantly adopted by use case approach which is inherently the functional analysis modelling approach. AFMUC, shifts this focus to misuse cases, the most ideal for security threat modeling and mitigation. Also, it facilitates more precise and systematic analysis of cross-cutting security threats and corresponding mitigations using all essential AOM constructs such as pointcut, joinpoint, advice, its types, introduction and weaving. In addition, the AFMUC explicitly and precisely specifies the attack surfaces within the use case basic flow, precondition, postcondition and dependency, that are vulnerable to threats. Similarly, the AFMUC mitigation template significantly enhances mitigation analysis by precisely specifying the mitigation surfaces with the misuse cases precondition, basic threat flow and postcondition. It also proposes the use of introduction constructs and weaved the aspect threat model into base use case model and aspect mitigation into misuse case model in a novel manner. which is a more unambiguous and formalized approach to identify the vulnerable points in the use case model and to analyze their corresponding mitigations.

Table 10. Comparative analysis of (mis)use case features used in AOM constructs.

| Reference | Precondition | Basic flow | Alternate flow | Postcondition | Dependency |
|---|---|---|---|---|---|
| [10,14] | – | – | – | – | – |
| [15,16] | – | – | – | – | – |
| [17] | – | – | – | – | – |
| [18] | + | + | – | + | – |
| [19] | – | + | + | – | – |
| [20] | – | – | – | – | – |
| [21] | + | + | + | + | – |

**Comparison to existing UML based aspect oriented approaches.** Several use case profiles proposed in literature to address the crosscutting concerns [12,13,22–27] for different UML diagrams. Table 11 assesses the existing UML profiles focusing on modelling crosscutting security concerns. It characterizes these works with respect to their purpose, and coverage of requirements. For instance, [23,24,26,27] model crosscutting security requirements into UML models using stereotypes, constrain, tagged values, or annotations. Nevertheless, these methods usually depend on security annotations that are incorporated into UML models, which leads to tightly coupled with UML model reducing modularization. Also, they frequently result in scattered and ambiguous representations of crosscutting security concerns. Table 11 also suggests that UML profile in [13] provide implementation of AOM constructs, i.e., pointcut, joinpoint, advice, and weaving to model crosscutting security threats but it supports the modelling in interaction diagram, e.g., sequence diagram.

In comparison, the proposed AFMUC approach enhances modularization by separating the crosscutting security threats and mitigations from core functional requirements. It adopts an AOM based approach, allowing security threats and mitigations to be specified independently as aspect model and then woven into the functional model. This leads to more scalable and maintainable solutions for modeling crosscutting security concerns in complex systems. Additionally, AFMUC provides analysis of crosscutting security threats and mitigation early in the software requirements phase in SDLC, supporting secure design by facilitating transformation into aspect oriented behavioral models such as mal sequence diagram [13] and AspectSM [12].

**Comparison to existing threat modelling approaches.** In comparison to existing threat modeling approaches such as STIDE [42], DREAD [43] and Attack trees [44] provide valuable analyses to the security threats. Table 12 suggested that STRIDE [42] focuses on identifying and categorizing threats and utilizes Data Flow Diagrams (DFDs) to model system boundaries, events, and entities. Another threat analysis approach DREAD [43] is a risk assessment and prioritization model used to evaluate threats. It emphasizes prioritization and mitigation using class-specific controls.

**Table 11. Comparative analysis of existing AOM based UML profiles.**

| Reference | Approach | Crosscutting Requirements | Purpose |
|---|---|---|---|
| [22] | Secure UML | Security policies | Integrate role-based Access Control RBAC security policies into UML models |
| [23,24] | UMLSec | Security requirements | Stereotype, constraint, and tagged value to annotate security with a base UML model |
| [25] | SECTECT | Security policies | Access control security policies are integrated with SOA |
| [26] | ModelSec | Security requirements | used model annotation, transformation, and mapping to integrate security requirements with the base model at different levels of abstraction in model-driven architecture |
| [27] | SecureMDD | Security requirement | Security annotations used with base UML models are used in automated code generation |
| [12] | AspectSM | Robustness behaviour | To model crosscutting robustness behavior based on different events using pointcut, joinpoint, advice, introduction and weaving |
| [13] | Aspect sequence diagram (mal sequence) | Security requirements | To model how crosscutting threats and mitigation interact with the base sequence diagram using pointcut, joinpoint, advice and weaving. |

**Table 12. Comparative analysis of existing threat modelling approaches.**

| Threat modelling technique | Purpose | Model | Crosscutting security concerns |
|---|---|---|---|
| STRIDE [42] | Identify and classify threats into categories (Spoofing, Tampering, etc.) | Data Flow Diagram (DFD) | Not supported |
| DREAD [43] | Quantify and prioritize risks associated with threats | Risk Assessment Matrix | Not supported |
| Attack Trees [44] | Provide hierarchical decomposition of attack goals and methods | Tree diagram (nodes, branches) | Not supported |

Attack Trees [44] provide a hierarchical representation of attack goals and the different ways to achieve them. Table 12 provides that these approaches work well within their scope, they are particularly used to detect threats, priorities threats, and give structural view of threat.

The AFMUC approach provides precise and formalized modeling of crosscutting threats behavior, interaction of threats with the core functional requirements while identifying the attack surfaces. Additionally, it also supports mitigation analysis by identifying crosscutting mitigations, analyzing their behavior, and understanding their interaction with corresponding threats while determining appropriate mitigation surfaces. AFMUC is important for security engineers to understand threat behavior, locate vulnerabilities and mitigation points in the model for secure application development.

**Limitation.** In use case specification template, alternate flow is a fundamental element to describe the deviation of the flow from basic successful scenario of use case. Alternate flow can also expose to potential vulnerabilities to be attacked by crosscutting security threats. Although the proposed AFMUC method effectively finds attack surfaces in Usecase basic flows, postconditions, and preconditions, it may not be suitable in the scenario involving alternate flow. This gap could potentially affect overall system security by making it more difficult to identify and modularize crosscutting threats that appear specifically within these alternate flows. Moreover, mitigation surfaces corresponding to alternate threat flows in misuse cases also require careful identification and modelling to ensure robust mitigation mechanism. Extending AFMUC to explicitly incorporate alternate flows in use cases and alternate threat flow in misuse case is crucial. This extension would enable the identification and modularization of crosscutting threats and their corresponding mitigation surfaces throughout all execution paths, enhancing the AFMUC ability to support comprehensive security threat and mitigation analysis.

## Threats to validity

This section describes the threats to validity of the experiment conducted to evaluate the proposed AFMUC approach, following the guidelines by Wohlen et al. [38].

### Conclusion validity

One of the potential threats to the proposed approach is that the generalizability of the results depending on the sample size selected for the experiment is a good representative of the students. To reduce this threat, a convenience sample technique [40] is followed to select students who are available at the moment and meet the criteria of the study. To generalize the results, the findings are also statistically analyzed using a non-parametric test, specifically the Spearman correlation test which is suitable for the small sample size.

### Internal validity

Another threat to the validity of results relates to the experiment design and procedure. To limit these threats, the experiment is piloted with the undergraduate students and faculty of the information security program to reduce any potential bias and ambiguity in the design of experiment objects. At the beginning of the experiment comprehensive training on the new approach aspect-oriented modelling and AFMUC approach is provided to the participants. Additionally, participants are given a home-based assignment on an example case study, i.e., EU-Rent car. Before the execution of the experiment, feedback is provided to individual participants. The assignment is graded to keep participants motivated to learn. Additionally, to assess the results that are based on subjective opinions, participants are asked to complete a quiz during the experiment to prevent potential bias while applying the proposed approach. The quiz is used to check the consistency of the results obtained from experiment and quiz but is not used to conclude the research outcome. Additionally, to maintain the authenticity and integrity of results, the participants are graded in their course while participating in experiment.

## Construct validity

There is a threat of the case study design to cover the proposed approach. To validate the AFMUC approach, MFA case study is designed, encompassing sufficient use cases threatened by aspect threats and their aspect mitigation. Five aspect threats threatening two use cases are selected along with one mitigation to mitigate these aspect threats. This ensures comprehensive coverage of the restriction rules, this selection included coverage of all essential constructs of the proposed approach such as pointcut, joinpoint, advice, and introduction. The small excerpt of case study is deliberately chosen because the primary objective is to evaluate the initial viability and coverage of the approach within a specific and manageable context. However, it is important to note that, to the best of current knowledge, the approach presented is novel.

## External validity

There is a threat to the validity of time and selection of students as subjects instead of professionals. This threat is addressed by keeping the industrial case study small, including three substantial use cases, five aspect threats, and one mitigation. The focused timeframe of three hours is thoughtfully chosen for a session to effectively assess the AFMUC constructs and restriction rules. This duration is sufficient to guide participants through the tasks without causing exhaustion while allowing for the thorough application and assessment of AFMUC. That is enough to complete in a three-hour session and cover the proposed approach. Selecting students as participants has posed another threat. However, students with suitable knowledge, skills, and backgrounds are chosen, and they have also received comprehensive training on the approach. This decision is made because industrial experts and practitioners have limited knowledge of AOM and would require significant training, time, and cost to learn it. Conversely, students are more inclined to learn new concepts.

## Conclusion

This paper presents a misuse case driven aspect-oriented approach called AFMUC to model the crosscutting security threats and their mitigations. The motivation is to develop secure online software applications by addressing and modelling the crosscutting security threats early in the software development phase ideally in the requirements analysis phase.

The proposed AFMUC approach has adopted the existing misuse case and security use case templates and extended them with aspect-oriented modeling constructs and restriction rules. The key properties of AFMUC modelling approach include specification of the crosscutting threats and crosscutting mitigations in an unambiguous and formalized manner. It enables precise analysis of aspect threats by identifying multiple attack surfaces in use cases, and the aspect mitigations to mitigate multiple threats by identifying mitigation surfaces in misuse cases. The aspect threats and mitigation specification are formalized using AFMUC restriction rules covering essential constructs of aspect-oriented modelling such as pointcut, joinpoint, advice, and introduction. The proposed approach is empirically evaluated with 24 students in a controlled environment, having background knowledge of information security. The ambiguity of AFMUC is assessed in terms of its understandability, applicability, restrictiveness, and error rate. The results show that the AFMUC constructs, and restriction rules are understandable and applicable with low restrictiveness. These findings have also been shown to be statistically significant. However, in terms of error rate, the results are insignificant. The reason for such results may be the aspect-oriented approach and use of restriction rules are new to subject. The analysis of the study summarizes that the proposed approach is novel and effectively addresses the challenge of modeling crosscutting security threats and their mitigation by utilizing misuse cases and mitigation cases early in the software requirements phase. While the proposed approach shows considerable promise in addressing security concerns through aspect-oriented modeling, there is still a significant scope for improvement. One key area for enhancement is extending support to additional elements of use cases, such as alternate flows and exception scenarios, which are currently not addressed. These elements can contain critical security-relevant behaviors and potential vulnerabilities that need to be modeled and analyzed effectively. Future work could focus on identifying join points within these alternate and exceptional flows to enable comprehensive threat

modeling. Additionally, improvements could include developing tool support to facilitate automation, enhance usability, and encourage wider adoption in industrial settings.

## Future work

In future work, one of the directions is to conduct extensive empirical evaluation of the AFMUC approach using larger sample sizes and real-world case studies. This will allow us to assess the scalability, adaptability, and practical impact of the framework in diverse application domains. These experiments will also help validate the generalizability and robustness of the proposed AFMUC approach against crosscutting security threats and mitigations. We plan to formally verify the AFMUC approach using formal methods and automate the weaving of the aspect model into the base model. Efforts can focus on developing a tool to identify cross-cutting security threats and corresponding mitigations using AFMUC, to enhance its adoption and practicality for practitioners. Furthermore, collaborating with industry partners to replicate the study in real-world settings and comparing the results with student data will further demonstrate efficiency and applicability of the approach. More studies can be conducted to further evaluate the consistency of proposed approach in transforming to other analysis models, code generation, and testing for secure software development.

## Supporting information

**S1 Appendix. Quiz Questionnaire.**
(PDF)

**S2 Data. Raw Data.**
(RAR)

## Author contributions

**Conceptualization:** Shumaila Iqbal, Rizwan Bin Faiz.

**Data curation:** Shumaila Iqbal.

**Investigation:** Shumaila Iqbal, Rizwan Bin Faiz, Muhammad Usman.

**Methodology:** Shumaila Iqbal, Muhammad Usman.

**Supervision:** Rizwan Bin Faiz.

**Validation:** Shumaila Iqbal.

**Writing – original draft:** Shumaila Iqbal.

**Writing – review & editing:** Rizwan Bin Faiz, Muhammad Usman, Shafiq Ur Rehman.

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
