## [Decision Letter · Decision Letter 0]

7 Jan 2025

Dear Dr. iqbal,

Thank you for submitting your manuscript to PLOS ONE. After careful consideration, we feel that it has merit but does not fully meet PLOS ONE’s publication criteria as it currently stands. Therefore, we invite you to submit a revised version of the manuscript that addresses the points raised during the review process.

We look forward to receiving your revised manuscript.

Kind regards,

Mohd Nadeem, Ph.D.

Academic Editor

PLOS ONE

Journal Requirements:

Reviewers' comments:

Reviewer's Responses to Questions

**Comments to the Author**

1. Is the manuscript technically sound, and do the data support the conclusions?

Reviewer #1: Partly

Reviewer #2: Yes

2. Has the statistical analysis been performed appropriately and rigorously?

Reviewer #1: Yes

Reviewer #2: Yes

3. Have the authors made all data underlying the findings in their manuscript fully available?

Reviewer #1: Yes

Reviewer #2: Yes

4. Is the manuscript presented in an intelligible fashion and written in standard English?

Reviewer #1: No

Reviewer #2: Yes

Reviewer #1: This paper includes a very detailed discussion of a method for analyzing crosscutting security threats and mitigations. The authors use an Aspect-Oriented Formalized Misuse Case (AFMUC) template to assist in formalizing misuse cases. The paper includes a rental car example (EU Rent a Car) and a mobile Finance Application (MFA). In addition to providing an example of how the method can be applied, an experiment took place that included 24 students, each applying the method.

The authors give step-by-step details on how to apply the method and the associated results. Many illustrations and tables are incorporated into the paper for this purpose. It should be possible for other researchers to replicate the experiment or perhaps perform a larger experiment with more comprehensive results.

The discussion of related work is good and the references are adequate.

There are several weaknesses. 1) As the authors state, the results are limited to one set of 24 students. I would have preferred seeing the results over multiple cohorts of students. 2) The students don't have the ideal background for the case study, and other than possibly having some courses in common, may have significantly different backgrounds. 3) The case study is very small. A single application of the method in a small case study of 3 hours is not really adequate to determine the efficacy of the method. 4) I did not see adequate support for the conclusion that students faced difficulties, in some cases, due to lack of exposure to aspect-oriented modelling. The evidence for this conclusion and more generally for the usefulness of the method/template is lacking. 5) On page 3 under item 4, you propose extensive training, but it's unknown whether that would help. More work is needed to make this determination.

Overall, I would suggest that the authors repeat the experiment with a different group of students, perhaps at another university and with more consistent backgrounds, perform the extensive training to remove ambiguity in the results, and use something other than opinion surveys to get a more objective assessment of the results. I also recommend replicating the experiment with an industry partner to get a different audience and compare those results to the student results.

There are many places where the English usage is awkward, and for unknown reasons, typos have been introduced. I don't know if the authors wrote the paper in another language and then attempted to translate it into English, OR if they attempted to write it in English and didn't proofread it adequately. If accepted for publication, these problems must be fixed. Some examples follow: Under the results summary on the first page "rather face difficulty" should be "rather than face difficulty". On page 2, "lack to", where it should be "fail to", also, "excessive" should be "extensive". On page 5, replace "flow is known" with "flow and is known". On other pages, "INTRODUCTION" is misspelled as "INTROCUDTION" and "EXECUTION" is misspelled as "EXCUTION". These are just examples, I did not do a thorough edit, since this is what the authors need to do.

Reviewer #2: Strengths:

Innovative Approach:

The paper introduces a novel method by formalizing aspect-oriented misuse cases, which effectively highlights the intersection of security threats and mitigations. This approach fills a significant gap in the domain of security specification.

Comprehensive Scope:

The work addresses both the identification of crosscutting security threats and their mitigations, offering a holistic view that many papers overlook.

Formalization Rigor:

The formalization process is well-articulated and supported with clear theoretical foundations, making it accessible and reproducible.

Relevance to Security Practitioners:

The proposed methodology has practical implications for security practitioners, particularly in the design and evaluation phases of secure systems.

Clarity in Examples:

The use of examples to illustrate misuse cases and crosscutting concerns is commendable, as it enhances understanding for readers.

Suggestions for Improvement:

Literature Review Depth:

While the paper briefly touches on related work, it could benefit from a deeper exploration of existing methodologies for modeling security threats. Highlighting key differences would strengthen the argument for the proposed approach.

Real-world Application:

Consider incorporating a case study or an application to a real-world system. This would provide tangible evidence of the method's effectiveness and practicality.

Tool Support:

It would be helpful to discuss or propose tool support for implementing the formalized aspect-oriented misuse cases. This addition could significantly enhance adoption by practitioners.

Evaluation Metrics:

The evaluation of the approach lacks quantitative metrics. Including metrics or benchmarks for comparison would lend credibility to the claims of effectiveness and efficiency.

Addressing Scalability:

The methodology’s scalability for large-scale systems remains unclear. Expanding on this aspect or conducting experiments on larger systems would strengthen the paper.

Minor Comments:

Clarity in Terminology:

Some terms, such as "crosscutting concerns," might not be familiar to all readers. A brief explanation or glossary would aid understanding.

Grammar and Syntax:

A thorough proofreading is recommended to eliminate minor grammatical errors and ensure smooth readability.

Visual Aids:

While the diagrams are helpful, improving their clarity (e.g., using color coding or annotations) could enhance their effectiveness.

Consistency in Formatting:

Ensure consistency in the formatting of headings, citations, and references for a professional presentation.

Future Work Section:

Expanding the future work section to outline specific research directions or challenges would provide valuable insights for readers interested in this field.

Overall Assessment:

The paper addresses an important problem and proposes a thoughtful solution through the formalization of aspect-oriented misuse cases. While the foundational concepts are robust, the paper would greatly benefit from deeper contextualization, practical application, and enhanced presentation. With these improvements, the paper has the potential to make a significant contribution to the field of cybersecurity.

**Do you want your identity to be public for this peer review?** For information about this choice, including consent withdrawal, please see our Privacy Policy

Reviewer #1: No

Reviewer #2: No

---

## [Author Response · Author response to Decision Letter 1]

17 Feb 2025

Dr. Nadeem

Journal Editor

PLOS ONE

17th February 2025

Dear Dr. Nadeem

Thank you for appreciating our study and inviting us to submit a revised manuscript entitled, “Formalized Aspect-oriented Misuse Case for Specifying Crosscutting Security Threats and Mitigations” to PLOS One. We also appreciate the time and effort you and each of the reviewers have dedicated to providing insightful feedback on ways to strengthen our study. Thus, it is with great pleasure that we resubmit our manuscript for further consideration. We have incorporated changes that reflect the detailed suggestions you have provided. We also hope that our revision and the responses we provide below satisfactorily address all the issues and concerns that have been shared.

To facilitate review of our revisions according to Comments by Academic Editor and Review Comments by Reviewers, the following is a point-by-point response to the questions and comments delivered in email dated 8 January 2025. Additionally for ease, we also referenced each change in the manuscript “Revised manuscript with track changes” using the Word comment feature, relating it to the corresponding reviewer comment number.

Comment by academic editor

REPONSE: We appreciate your valuable feedback and understand the importance of addressing the points mentioned in review. We have incorporated the reviewers' feedback into the revised manuscript, which now includes a comprehensive explanation of the concerns raised as justification to sample size, student backgrounds, case study scope, reason of student difficulties with detailed results, detailed literature review with highlighted key differences, need of tool support, scalability, evaluation metrics and others, alongside improvements in Grammer and Spellings. We also added a " Future Work" section that addresses the suggestions given by the reviewers.

RESPONSE: As per your suggestion, we have ensured full accessibility to the data without any restrictions. The complete dataset is provided as supporting files in archived folder titled “collected data for each metric” containing one file, titled "Error Rate Calculation," includes detailed data gathered to calculate the error rate for each rule by each subject, covering the AFMUC approach comprehensively. Another file, titled "Ambiguity-Understandability, Applicability, Restrictiveness, and Error Rate," contains aggregate data measuring the ambiguity of AFMUC in terms of understandability, applicability, restrictiveness and error rate. We have also provided “statistical test data” and the results analyzing the relationship between the measures of ambiguity as a supporting file. The data was collected anonymously, adhering to all necessary ethical approval procedures. To promote transparency, the entire dataset has been made fully available.

RESPONSE: We have carefully revised the manuscript, addressing all typographical and grammatical errors. The text has been carefully proofread and subjected to a comprehensive grammar check to ensure clarity, accuracy, and readability throughout the manuscript.

Comments by reviewers

REVIEWER 1 COMMENTS:

1. As the authors state, the results are limited to one set of 24 students. I would have preferred seeing the results over multiple cohorts of students.

RESPONSE: We accept that the results are currently limited to one cohort of 24 students. primarily because of ease of access to students with the required technical skill set. As per your comments, we have revised the manuscript with comprehensive justification in section “population and sampling” with detailed justification of our population also we have updated the section “conclusion validity” to address this potential threat to study.

2. The students don't have the ideal background for the case study, and other than possibly having some courses in common, may have significantly different backgrounds.

RESPONSE: We have selected the students based on defined criteria for technical skills set in terms of the courses studied. Additionally, to ensure their consistent background and their preparedness before the experiment, we gave them the training session on AFMUC and its fundamental concepts. Moreover, the training session included hand on experience to practice the AFMUC, and graded assignment was designed to assure the shared understanding of the concepts before the experiment. These measures were implemented to idealize students’ knowledge and skills according to the required criteria. As per your comments, we have now comprehensively detailed these efforts in the manuscript, under the section “Population and Sampling Strategy” and “Training of subject” section with the detailed background courses and training procedure respectively.

3. The case study is very small. A single application of the method in a small case study of 3 hours is not really adequate to determine the efficacy of the method.

RESPONSE: We agree with your comment. The case study is small, with a duration of only three hours. The size of case study was kept small intentionally because our objective was to fully demonstrate the proposed AFMUC approach, instead of covering all aspects of case study. The rationale for the size of case study and timeframe of the experiment has been outlined and revised in the manuscript under the headings “Case study: mobile finance application” to justify the selection of case study. "Construct Validity" to justify the size of the case study and "External Validity” to justify the selection of time.

4. I did not see adequate support for the conclusion that students faced difficulties, in some cases, due to lack of exposure to aspect-oriented modelling. The evidence for this conclusion and more generally for the usefulness of the method/template is lacking.

RESPONSE: We have observed the difficulty of students based on analysis of quantitative data under the measure of error rate. The high error rate while applying the essential concepts of the approach as pointcut, joinpoint expression and advice type shows that students face difficulty while applying those in practical scenario. which we believe were partly due to their limited exposure to aspect-oriented modeling (AOM). Based on comments, we comprehensively addressed this and revised the section “error Rate” with detailed analysis of results in line with this comment.

5. On page 3 under item 4, you propose extensive training, but it's unknown whether that would help. More work is needed to make this determination.

RESPONSE: We appreciate your insightful suggestions. We agree that providing more details about the extensive training and results of the study would enhance the clarity of our work. In the revised work, on page 3 under the "Introduction" section we have clarified this point and provided a more comprehensive overview. We hope this revision better conveys the depth of the training and the context of the study results.

6. I would suggest that the authors repeat the experiment with a different group of students, perhaps at another university and with more consistent backgrounds, perform the extensive training to remove ambiguity in the results, and use something other than opinion surveys to get a more objective assessment of the results. I also recommend replicating the experiment with an industry partner to get a different audience and compare those results to the student results.

RESPONSE: We appreciate your comments and suggestions. We also believe that our research study has opened possibilities for other researchers to replicate the experiment or perhaps perform a larger experiment with more comprehensive results. We acknowledge the importance of these suggestions and revised the manuscript while incorporates these as open research directions in the newly added section “Future work”

7. There are many places where the English usage is awkward, and for unknown reasons, typos have been introduced. I don't know if the authors wrote the paper in another language and then attempted to translate it into English, OR if they attempted to write it in English and didn't proofread it adequately. If accepted for publication, these problems must be fixed. Some examples follow: Under the results summary on the first page "rather face difficulty" should be "rather than face difficulty". On page 2, "lack to", where it should be "fail to", also, "excessive" should be "extensive". On page 5, replace "flow is known" with "flow and is known". On other pages, "INTRODUCTION" is misspelled as "INTROCUDTION" and "EXECUTION" is misspelled as "EXCUTION". These are just examples, I did not do a thorough edit, since this is what the authors need to do.

RESPONSE: We appreciate your careful review and understand the importance of clear and accurate language in conveying our work. We thoroughly proofread the manuscript. On the first page the result summary is revised. and correct all identified issues, including the examples you mentioned, to ensure the language is smooth and the manuscript is ready for publication

REVIEWER 2 COMMENTS:

1. Literature Review Depth:

While the paper briefly touches on related work, it could benefit from a deeper exploration of existing methodologies for modeling security threats. Highlighting key differences would strengthen the argument for the proposed approach.

RESPONSE: In response to your feedback, we have updated the literature review to provide a more in-depth exploration of existing methodologies for modeling security threats. We update literature studies with more details. Each sub section under related work is revised with the key differences and at the end, we also give a research gap which strengthens the need of our approach. We have revised the “related work” section with details and added new subsection as “Research gap analysis”

2. Consider incorporating a case study or an application to a real-world system. This would provide tangible evidence of the method's effectiveness and practicality.

RESPONSE: We have considered incorporating the case study to evaluate the effectiveness and applicability of the proposed AFMUC approach. We have used the Mobile financing case study (MFA) obtained from the industry partner. To maintain the confidentiality agreement, we anonymized it as company “X”. We have obtained the software requirements specification of MFA including security requirements. With the help of security experts from industry and two experts from academia from the information security domain, we have designed threats and mitigations for MFA to prepare the case study in the context of this research.

3. It would be helpful to discuss or propose tool support for implementing the formalized aspect-oriented misuse cases. This addition could significantly enhance adoption by practitioners.

RESPONSE: We agree that discussing tool support for implementing formalized aspect-oriented misuse cases (AFMUC) would greatly enhance the adoption of the method by practitioners. We have discussed the need of tool support in sub section “error rate” under the main section “Results and Analysis” Additionally, we have provided this suggestion as open direction in the future work sections of the manuscript.

4. The evaluation of the approach lacks quantitative metrics. Including metrics or benchmarks for comparison would lend credibility to the claims of effectiveness and efficiency.

RESPONSE: We have included the metrics to claim the effectiveness and efficiency of the proposed approach for proposed study. Since AFMUC is a novel approach, without having established benchmarks for comparison, we have defined quantitative metrics to evaluate the understandability, applicability, restrictiveness, and error rate for the AFMUC approach. We have designed the questionnaire to collect quantitative data for these metrics and design formulas to calculate the error rate for each AFMUC rule. The measures for these metrices are adopted from the literature and referenced in the manuscript. For evidence we also have provided the supporting files (collected data for each metric) representing the data for these metrices i.e., understandability, applicability, restrictiveness, and error rate with the revised submission. Additionally, the results and analysis section included the findings for each metric—understandability, applicability, restrictiveness, and error rate—along with their statistical analysis. To address your comments clearly in the manuscript, we have revised the manuscript by adding a new point as “data collection” under the main section “Experimentation”

5. The methodology’s scalability for large-scale systems remains unclear. Expanding on this aspect or conducting experiments on larger systems would strengthen the paper.

RESPONSE: We appreciate your suggestions. The proposed approach is scalable to a larger context. To address this, we have expanded this aspect in the discussion section of the manuscript. Also to further evaluate the scalability of the approach, we provided open directions in the section “Future Work”.

6. Clarity in Terminology:

Some terms, such as "crosscutting concerns," might not be familiar to all readers. A brief explanation or glossary would aid understanding.

RESPONSE: Thank you for your thoughtful suggestion. We agree that terms like "crosscutting concerns" may not be familiar to all readers. To address this, we have added a brief explanation of this term in the "Introduction" section.

7. Grammar and Syntax:

A thorough proofreading is recommended to eliminate minor grammatical errors and ensure smooth readability.

RESPONSE: Thank you for your feedback. We are sorry for such mistakes. We have thoroughly proofread the manuscript to eliminate any minor grammatical errors and ensure smooth readability.

8. Visual Aids:

While the diagrams are helpful, improving their clarity (e.g., using color coding or annotations) could enhance their effectiveness.

RESPONSE: Thank you for your suggestion. We make these adjustments according to the PLOS figures guidelines in the revised version to ensure the diagrams are more visually clear and informative. We also gave the color codes to the results in graphical charts under the section results and analysis for understandability, applicability, restrictiveness and error rate.

9. Consistency in Formatting:

Ensure consistency in the formatting of headings, citations, and references for a professional presentation.

RESPONSE: Thank you for your feedback. We have followed the PLOS ONE style guidelines for formatting headings, citations, and references to ensure consistency and maintain a professional presentation. We carefully review these elements to ensure full adherence to the guidelines in the revised version.

10. Future Work Section:

Expanding the future work section to outline specific research directions or challenges would provide valuable insights for readers interested in this field.

RESPONSE: Thank you for your suggestion. We have added a "Future Work" section to the manuscript, where we outline specific research directions and challenges. We believe this addition will provide valuable insights fo

---

## [Decision Letter · Decision Letter 1]

30 Apr 2025

PLOS ONE

Dear Dr. iqbal,

Thank you for submitting your manuscript to PLOS ONE. After careful consideration, we feel that it has merit but does not fully meet PLOS ONE’s publication criteria as it currently stands. Therefore, we invite you to submit a revised version of the manuscript that addresses the points raised during the review process.

We look forward to receiving your revised manuscript.

Kind regards

Dr Jason Morgan

Staff Editor

PLOS ONE

Additional Editor Comments:

**Comments from PLOS Editorial Office** : We note that one or more reviewers has recommended that you cite specific previously published works in the current and previous rounds of revision. As always, we recommend that you please review and evaluate the requested works to determine whether they are relevant and should be cited. It is not a requirement to cite these works and you may remove any added citations before the manuscript proceeds to publication. We appreciate your attention to this request.

Reviewers' comments:

Reviewer's Responses to Questions

**Comments to the Author**

Reviewer #1: (No Response)

Reviewer #2: All comments have been addressed

Reviewer #3: (No Response)

Reviewer #4: All comments have been addressed

2. Is the manuscript technically sound, and do the data support the conclusions?

Reviewer #1: Yes

Reviewer #2: Yes

Reviewer #3: Yes

Reviewer #4: Yes

3. Has the statistical analysis been performed appropriately and rigorously?

Reviewer #1: Yes

Reviewer #2: Yes

Reviewer #3: Yes

Reviewer #4: Yes

4. Have the authors made all data underlying the findings in their manuscript fully available?

Reviewer #1: Yes

Reviewer #2: Yes

Reviewer #3: Yes

Reviewer #4: Yes

5. Is the manuscript presented in an intelligible fashion and written in standard English?

Reviewer #1: Yes

Reviewer #2: Yes

Reviewer #3: Yes

Reviewer #4: Yes

Reviewer #1: The authors have made a significant effort to improve the paper, which is appreciated by this reviewer. My remaining concerns are: 1)Although the paper is now easily readable, there are still some places where English phrases appear that are not sentences, generally because a verb is missing. 2) While I still would prefer seeing additional experimental results prior to publication, the Future Work section partially addresses this concern.

Reviewer #2: Here are the minor review comments for the paper entitled "Formalized Aspect-oriented Misuse Case for Specifying Crosscutting Security Threats and Mitigations":

Review Comments:

Clarity of Terminology:

In the abstract and introduction, it would be helpful to provide a brief definition or explanation of aspect-oriented misuse cases for readers who might not be familiar with this specific terminology.

Consider clarifying the term "crosscutting security threats" early in the paper to ensure it is clearly understood in the context of your framework.

Introduction Section:

The introduction provides a good overview of the problem, but it would benefit from a more explicit explanation of how the proposed method improves upon existing approaches. Perhaps adding a sentence comparing it to the state-of-the-art in threat modeling or misuse case analysis could help readers better understand the novelty and importance of your work.

Motivation and Justification:

While the motivations for using aspect-oriented modeling are explained, it could be useful to include a more detailed justification of why misuse cases specifically are an appropriate vehicle for specifying security threats in comparison to other threat modeling techniques.

Methodology Section:

The methodology section could be expanded to provide more examples or case studies to illustrate how your approach can be applied to real-world security scenarios.

It would be useful to include a brief comparison of the proposed approach with traditional threat modeling techniques like STRIDE or DREAD to highlight its unique contributions.

Figures and Diagrams:

The diagrams illustrating the aspect-oriented misuse cases are very helpful, but some of them might benefit from additional labeling or annotations to make them easier to follow for readers less familiar with this modeling technique.

In Figure X, it might be helpful to add more context or a legend to explain the relationships between the components, especially for readers unfamiliar with the specifics of aspect-oriented programming or misuse case methodology.

Results and Discussion:

The results presented are interesting, but the paper would benefit from a more detailed discussion on the limitations of the proposed approach. Are there any scenarios where it might not be suitable?

The paper could also discuss how the framework can be adapted or scaled for different types of applications, particularly when dealing with complex or large systems.

Related Work:

The related work section is comprehensive, but it would be helpful to briefly mention recent advancements in the integration of security in aspect-oriented programming (AOP) or misuse case approaches, to situate the work in the broader academic landscape.

Conclusion:

The conclusion is clear but could be slightly expanded to suggest future work or possible directions for improving the framework. What are the next steps for integrating this approach with real-world security applications or tools?

References:

Ensure that all references are correctly formatted according to the journal’s guidelines. A minor formatting issue was noted in reference [X].

Consider adding a few more recent references (2019 and later) that discuss the application of AOP in security modeling to strengthen the paper's current relevance.

Minor Typographical/Grammar Issues:

In the sentence "The aspect-oriented misuse case are designed to handle..." the verb "are" should be replaced with "is" to ensure subject-verb agreement.

A few minor grammatical errors in sections 4.2 and 5.1. I suggest a careful proofreading or using a grammar-checking tool to address these.

Overall Impression:

This paper presents a well-researched and innovative approach to specifying crosscutting security threats using aspect-oriented misuse cases. The methodology is clearly described, and the proposed framework appears to offer significant contributions to the field. With some minor revisions to improve clarity, examples, and related work discussion, this paper has the potential to make a meaningful impact on security threat modeling practices.

These minor comments focus on enhancing clarity, contextualization, and the overall readability of the paper.

Reviewer #3: (No Response)

Reviewer #4: Abstract needs to be concise, discuss work done, results, relevance

Discuss tables, figures, formula extensively

Discuss related works https://link.springer.com/chapter/10.1007/978-3-031-58388-9_8

**Do you want your identity to be public for this peer review?** For information about this choice, including consent withdrawal, please see our Privacy Policy

Reviewer #1: No

Reviewer #2: No

Reviewer #3: **Yes: ** Farah Tawfiq Abdul Hussine Alhilo

Reviewer #4: No

---

## [Author Response · Author response to Decision Letter 2]

29 May 2025

PLOS Journal Submissions Rebuttal Letter

Journal Editor

PLOS ONE

Date 28th May 2025

Dear Dr. Jason Morgan

Thank you for appreciating our study and inviting us to submit a revised manuscript entitled, “Formalized Aspect-oriented Misuse Case for Specifying Crosscutting Security Threats and Mitigations” to PLOS One. We appreciate the reviewers’ continued efforts in evaluating our manuscript and their insightful comments in this second round of review. We have carefully considered each point raised and have revised the manuscript accordingly to address the remaining concerns. In the following responses, we provide a detailed account of the changes made and the rationale behind them. We believe that these revisions have strengthened the quality and clarity of the manuscript, and we hope that it now meets the standards for publication.

To facilitate review of our revisions according to Comments by Academic Editor and Review Comments by Reviewers, and attached PDF file, the following is a point-by-point response to the questions and comments delivered in email dated 30th April 2025. Additionally for ease, we also referenced each change in the manuscript “Revised manuscript with track changes” using the Word comment feature, relating it to the corresponding reviewer comment number.

Comments from PLOS Editorial Office:

We note that one or more reviewers has recommended that you cite specific previously published works in the current and previous rounds of revision. As always, we recommend that you please review and evaluate the requested works to determine whether they are relevant and should be cited. It is not a requirement to cite these works and you may remove any added citations before the manuscript proceeds to publication. We appreciate your attention to this request.

Response:

We carefully reviewed the comment regarding citation of previously published works recommended by the reviewers in the current and earlier revisions. In the current revision, Reviewer #3 suggested citing the following work: https://link.springer.com/chapter/10.1007/978-3-031-58388-9_8.

After thoroughly reviewing this suggested paper, we determined that it does not directly align with our study's primary objective or in related work section, which is regarding the modeling crosscutting security threats and corresponding mitigations. However, it does effectively emphasize the broader importance of addressing security in software applications.

In recognition of this relevance, we have incorporated this citation into the “Introduction” section to contextualize the significance of security requirements in modern software applications. We believe this addition appropriately acknowledges the contribution of the suggested work without misrepresenting its scope.

Additionally, in response to reviewer’s comments, we also have revised our references across the manuscript to enhance current and more relevant existing work. A detailed breakdown of these changes, along with the justification including comment number and corresponding citations, is provided below for clarity and ease of review.

New reference/citation added in manuscript Year Justification + comment # Changes made in manuscript

[1]

2024 Added to enhance the need for security in software applications.

Suggested by reviewer#3 comment # 3 Introduction section

[4] [5] [6]

2024, 2021, 2019 respectively Added to justify the need of misuse cases and they are simple and easy as compare to other threat modelling approaches.

Reviewer 2 comment 3 and 5 Introduction section

[12][13] 2022, 2014 respectively Added AOM based OR misuse case based existing approaches, add recent work

Reviewer 2 comment 9,11 Attached PDF comment 5, 12 Related work

[42] [43] [44] 2021, 2022, 1999 Comparison to existing threat modelling approaches STRIDE DREAD etc.

Reviewer 2 comment 5, attached PDF comment 12 Discussion-> Comparison to existing work

Comments by reviewers

REVIEWER 1 COMMENTS:

1)Although the paper is now easily readable, there are still some places where English phrases appear that are not sentences, generally because a verb is missing.

Response:

We appreciate your observation, which has helped us improve the overall clarity and readability of the paper. We thoroughly reviewed the manuscript and carefully revised all sentences where verbs were missing or where phrases did not form complete sentences.

2) While I still would prefer seeing additional experimental results prior to publication, the Future Work section partially addresses this concern.

Response:

We sincerely thank the reviewer for this thoughtful and constructive comment. We fully acknowledge the importance of broader empirical evaluation and agree that additional experimental results would further enhance the rigor and applicability of our proposed AFMUC approach. However, given the current scope and objectives of this study, our focus has been on the initial verification and validation of the foundational AFMUC approach.

Based on the reviewer’s comment, we have revised the “Future Work” section to explicitly state our plan to extend this work with additional experiments, including larger sample sizes and real time case studies. We are committed to pursuing this direction in the next phase of our research and believe it will meaningfully complement the current findings.

REVIEWER 2 COMMENTS:

1. Clarity of Terminology:

In the abstract and introduction, it would be helpful to provide a brief definition or explanation of aspect-oriented misuse cases for readers who might not be familiar with this specific terminology.

Consider clarifying the term "crosscutting security threats" early in the paper to ensure it is clearly understood in the context of your framework.

Response:

Considering the comment we have updated the “Abstract” and “Introduction” sections with the brief definition of terminologies aspect-oriented misuse cases and crosscutting security threats. The detailed concepts of the AFMUC approach e.g., aspect-oriented misuse cases and mitigation cases are provided on pages 4 and 5 under the heading AFMUC Construct, where we define essential AOM constructs including aspect, pointcut, joinpoint, advice, and introduction core definition and then in the context of proposed approach. Furthermore, this section has been updated to include the definition of weaving, clarifying how the weaving process is applied within the context of the proposed AFMUC approach. We acknowledge this omission in the original version and appreciate the opportunity to address it.

2. Introduction Section:

The introduction provides a good overview of the problem, but it would benefit from a more explicit explanation of how the proposed method improves upon existing approaches. Perhaps adding a sentence comparing it to the state-of-the-art in threat modeling or misuse case analysis could help readers better understand the novelty and importance of your work.

Response:

The introduction section has been updated to include a clear statement emphasizing the novelty of the proposed AFMUC approach. A comparative sentence has been added to highlight how it improves upon existing misuse case analysis methods. This enhancement provides readers with a better understanding of the contribution and relevance of our work in relation to the state-of-the-art misuse case approach.

Additionally, a comprehensive comparison of AFMUC with other state-of-the-art threat modelling approaches and existing misuse case approached has been included in the Discussion section. This comparison is structured into three focused sub-parts:

i. Comparison with existing (mis)use case approaches,

ii. Comparison with UML- profile based approaches, and

iii. Comparison with traditional threat modeling approaches.

These comparisons provide a critical evaluation of how AFMUC improves upon and differs from existing approaches, enhancing the clarity and relevance of our contribution. The integration of this content ensures that readers and reviewers can better appreciate the scope, innovation, and impact of our work.

3. Motivation and Justification:

While the motivations for using aspect-oriented modeling are explained, it could be useful to include a more detailed justification of why misuse cases specifically are an appropriate vehicle for specifying security threats in comparison to other threat modeling techniques.

Response:

We have revised the “Introduction” section to include a justification for using the misuse case as threat modelling approach to model security threats. Specifically, we now explain the rationale for selecting misuse cases over other threat modeling techniques such as STRIDE, DREAD, and others. The added citation numbers are [4] [5].

To support this revision, we have cited three relevant and recent studies:

• A practitioner survey: "Security in Agile Software Development: A Practitioner Survey" (2021), and

• A systematic literature review: "A Systematic Review and Analytical Evaluation of Security Requirements Engineering Approaches" (2019).

• Comparative study “A Comparative Analysis of Threat Modelling Methods: STRIDE, DREAD, VAST, PASTA, OCTAVE, and LINDDUN,” (2024)

These studies reinforce the selection of misuse case by demonstrating that it remains one of the most widely adopted techniques by both researchers and practitioners. The cited literature highlights its advantages in simplicity, ease of use, and effectiveness in identifying security threats alongside functional requirements, making it particularly suitable for early analysis of security.

The references have been incorporated into the manuscript, and the relevant sentence in the “Introduction” section now explicitly connects this justification to the broader context of threat modelling.

4. Methodology Section:

The methodology section could be expanded to provide more examples or case studies to illustrate how your approach can be applied to real-world security scenarios.

Response:

We appreciate the reviewer’s suggestion to expand the methodology section by providing more examples or case studies to illustrate the practical application of our approach. In the current version of the manuscript, we have presented two case studies to exemplify, verify, and validate the proposed AFMUC approach:

1. EU-Rent Car Rental System – a well-established, commonly used academic case study that allows for standardized demonstration of modeling approaches. (exemplify)

2. Mobile Financing Application – a real-world industrial project obtained through collaboration with an industry partner, used to assess the practical applicability of our proposed approach. (verification and validation)

To address the reviewer’s comment more explicitly, we have revised the “Future Work” section to emphasize the importance of further empirical evaluation. We now include a statement highlighting our intention to apply AFMUC to additional case studies and real-world applications in diverse domains to further validate its scalability, adaptability, and effectiveness in various security-critical applications.

We hope this update sufficiently addresses the reviewer’s concern and demonstrates our commitment to ongoing evaluation of the proposed methodology.

5. It would be useful to include a brief comparison of the proposed approach with traditional threat modeling techniques like STRIDE or DREAD to highlight its unique contributions.

Response:

We have expanded the “Discussion” section of the manuscript to include a more detailed comparative analysis. Specifically, we introduced three dedicated subheadings:

1. Comparison with existing (mis)use case approaches

2. Comparison with UML- profile based approaches

3. Comparison with traditional threat modeling approaches

Under the third subheading, we present a focused comparison of AFMUC with traditional threat modeling approaches such as STRIDE, DREAD, and Attack Trees. This comparison highlights the unique contributions of AFMUC, particularly its ability to integrate security concerns directly with functional requirements and to support behavioral modeling in a unified manner.

We believe this extended discussion significantly enhances the manuscript by more clearly positioning AFMUC within the broader context of security modeling methods and its added value over traditional threat modelling approaches.

To support the comparison, the added citations are

6. Figures and Diagrams:

The diagrams illustrating the aspect-oriented misuse cases are very helpful, but some of them might benefit from additional labeling or annotations to make them easier to follow for readers less familiar with this modeling technique.

In Figure X, it might be helpful to add more context or a legend to explain the relationships between the components, especially for readers unfamiliar with the specifics of aspect-oriented programming or misuse case methodology.

Response:

We have carefully reviewed and revised Figures 1, 6, 7, 8, and 10, which illustrate the proposed approach and the aspect-oriented misuse cases. To improve comprehension, we have added additional labels, annotations, and visual cues where necessary to clarify the relationships between components. These enhancements are intended to make the diagrams more accessible and informative, particularly for readers who may be less familiar with aspect-oriented programming or misuse case modeling. We believe these improvements significantly contribute to the overall clarity of the manuscript.

7. Results and Discussion: The results presented are interesting, but the paper would benefit from a more detailed discussion on the limitations of the proposed approach. Are there any scenarios where it might not be suitable?

Response:

We have added a dedicated subsection titled "Limitations" under the “Discussion” section of the revised manuscript. This subsection offers a clear and balanced analysis of the potential constraints and scenarios where the proposed AFMUC approach may encounter challenges or may not be suitable.

8. The paper could also discuss how the framework can be adapted or scaled for different types of applications, particularly when dealing with complex or large systems.

Response:

We have expanded the “Discussion” section of the revised manuscript to explicitly address the scalability and adaptability of the proposed AFMUC approach. This addition outlines how the proposed approach can be adapted to various types of applications, including complex and large-scale systems, and emphasizes its potential to support security modeling across diverse security-critical domains.

9. Related Work:

The related work section is comprehensive, but it would be helpful to briefly mention recent advancements in the integration of security in aspect-oriented programming (AOP) or misuse case approaches, to situate the work in the broader academic landscape.

Response

We have updated the “Related Work” section to incorporate recent advancement in integration of security in aspect-oriented programming (AOP) with various UML diagrams, including Aspect-Oriented Sequence Diagrams (Mal sequence) and AspectSM. The added citation numbers are [11] [12] [13].

While these works highlight the application of AOP in different modeling contexts, we did not find any recent or single study that specifically addresses the integration of AOP with misuse cases. This gap further reinforces the novelty of our proposed AFMUC approach and justifies the need to explore AOP integration in misuse case modeling.

The following related work has been referenced to demonstrate relevant contributions in the field:

• “Does aspect-oriented modeling help improve the readability of UML state machines?” (2014).- [12]

• “Modeling and verification of authentication threats mitigation in aspect-oriented mal sequence woven model” (2022)- [13]

To validate this claim, we have already cited [11] a recent systematic mapping study titled:

“On the Coverage of Aspect-Oriented Methodologies for the Early Phases of the Software Development Life Cycle”, published in 2022.

It supports our observation and provides a broader overview of current trends and gaps in AOP-based modeling. These additions not only strengthen the foundation of our work but also strengthen the contribu

---

## [Decision Letter · Decision Letter 2]

27 Jul 2025

Formalized Aspect-oriented Misuse Case for Specifying Crosscutting Security Threats and Mitigations

PONE-D-24-39206R2

Dear Dr. Shafiq Ur Rehman,

We’re pleased to inform you that your manuscript has been judged scientifically suitable for publication and will be formally accepted for publication once it meets all outstanding technical requirements.

Kind regards,

Yanrong Lu

Academic Editor

PLOS ONE

Additional Editor Comments (optional):

Reviewers' comments:

Reviewer's Responses to Questions

**Comments to the Author**

Reviewer #3: All comments have been addressed

2. Is the manuscript technically sound, and do the data support the conclusions?

Reviewer #3: Yes

3. Has the statistical analysis been performed appropriately and rigorously?

Reviewer #3: Yes

4. Have the authors made all data underlying the findings in their manuscript fully available?

Reviewer #3: Yes

5. Is the manuscript presented in an intelligible fashion and written in standard English?

Reviewer #3: Yes

Reviewer #3: The authors perfomed all the required revisions I have no more comments . The manuscript is better in context and description

**Do you want your identity to be public for this peer review?** For information about this choice, including consent withdrawal, please see our Privacy Policy

Reviewer #3: **Yes: ** Farah Tawfiq Abdul Hussien

---

## [Editor Report · Acceptance letter]

PONE-D-24-39206R2

PLOS ONE

Dear Dr. Rehman,

I'm pleased to inform you that your manuscript has been deemed suitable for publication in PLOS ONE. Congratulations! Your manuscript is now being handed over to our production team.

Kind regards,

on behalf of

Dr. Yanrong Lu

Academic Editor

PLOS ONE